# The extracellular matrix protects *Bacillus subtilis* colonies from *Pseudomonas* invasion and modulates plant co-colonization

Carlos Molina-Santiago [1], John R. Pearson [2], Yurena Navarro[1], María Victoria Berlanga-Clavero [1], Andrés Mauricio Caraballo-Rodriguez [3], Daniel Petras [3], María Luisa García-Martín [2], Gaelle Lamon[4], Birgit Haberstein [4], Francisco M. Cazorla [1], Antonio de Vicente [1], Antoine Loquet [4], Pieter C. Dorrestein [3] & Diego Romero [1]

Bacteria of the genera *Pseudomonas* and *Bacillus* can promote plant growth and protect plants from pathogens. However, the interactions between these plant-beneficial bacteria are understudied. Here, we explore the interaction between *Bacillus subtilis* 3610 and *Pseudomonas chlororaphis* PCL1606. We show that the extracellular matrix protects *B. subtilis* colonies from infiltration by *P. chlororaphis*. The absence of extracellular matrix results in increased fluidity and loss of structure of the *B. subtilis* colony. The *P. chlororaphis* type VI secretion system (T6SS) is activated upon contact with *B. subtilis* cells, and stimulates *B. subtilis* sporulation. Furthermore, we find that *B. subtilis* sporulation observed prior to direct contact with *P. chlororaphis* is mediated by histidine kinases KinA and KinB. Finally, we demonstrate the importance of the extracellular matrix and the T6SS in modulating the coexistence of the two species on melon plant leaves and seeds.

[1] Instituto de Hortofruticultura Subtropical y Mediterránea "La Mayora", Universidad de Málaga-Consejo Superior de Investigaciones Científicas (IHSM-UMA-CSIC), Departamento de Microbiología, Universidad de Málaga, Bulevar Louis Pasteur 31 (Campus Universitario de Teatinos), 29071 Málaga, Spain. [2] Nano-imaging Unit, Andalusian Centre for Nanomedicine and Biotechnology, BIONAND, 29071 Málaga, Spain. [3] University of California San Diego, Collaborative Mass Spectrometry Innovation Center, La Jolla, CA 92093, USA. [4] Institute of Chemistry & Biology of Membranes & Nanoobjects (UMR5248 CBMN), CNRS, Université Bordeaux, Institut Européen de Chimie et Biologie, 33600 Pessac, France. Correspondence and requests for materials should be addressed to D.R. (email: diego_romero@uma.es)

Plant-associated microorganisms can be divided into three groups, beneficial[1–4], deleterious[5–7], and neutral[8,9], depending on their effect on plant hosts. Bacteria belonging to these three groups continuously interact with their plant hosts and between themselves, constituting what is known as the plant microbiome[10]. Bacteria found in plant-related environments usually reside in structurally and dynamically complex biological systems known as biofilms, which can confer a range of benefits including increased resistance to environmental stresses, metabolic exchange, better surface colonization, and increased horizontal gene transfer[11,12]. Common to all bacterial biofilms is the presence of a secreted extracellular matrix that holds cells together and provides robustness to the biofilm architecture. Although extracellular matrix composition varies between species, in general it is composed of polysaccharides, proteins, and nucleic acids[13,14].

*Bacillus subtilis* is a soil-dwelling bacteria that live in harmony with plants and is used as a model of biofilm formation[14–17]. The extracellular matrix of *B. subtilis* is mainly composed of exopolysaccharides, synthesized by the *epsA-O* operon-encoded genes; TasA, a functional amyloid encoded in the three-gene operon *yqxM/tapA-sipW-tasA*[18,19]; and BslA, which is involved in the formation of a hydrophobic coat over the biofilm[20]. Although their role in biofilm assembly is well understood, little is known about the functional importance of extracellular matrix components in natural environments, such as on plant surfaces. Cells growing within a biofilm are in continuous contact with each other, as well as with their environment and other organisms living in the same ecological niche. Bacterial interactions are thus defined by a combination of different factors such as the activation of different metabolic pathways and the production and secretion of signaling compounds, siderophores, antibiotics, and quorum-sensing molecules[21–26]. These compounds can be excreted into the extracellular space by bacterial secretion systems, efflux pumps, transporters, and membrane vesicles. Secretion mechanisms can be compound-specific or be able to extrude a broad spectrum of compounds, with some requiring tight physical connections between interacting cells, such as the type VI secretion system (T6SS), a tubular puncturing device able to deliver molecules directly into other cells. For example, bacteria of the genus *Pseudomonas*, can use their T6SS to manipulate and subvert eukaryotic host cells and/or to fight other bacteria thriving in the same ecological niche[27,28].

*Pseudomonas* spp. and *Bacillus* spp. are among the most-predominant genera of plant-beneficial bacteria. Both genera have been well studied and their abilities to protect plants against pathogens[29–31] and to promote growth of many plant species have been widely described[32,33]. Examples are *B. subtilis* strain NCIB3610 (here referred to as 3610), a model organism characterized by its biocontrol properties and biofilm formation, and *Pseudomonas chlororaphis* PCL1606 (referred to as PCL1606), a strain isolated from the plant rhizosphere possessing antifungal activity[1,34–36]. However, studies examining how these bacterial species interact and co-exist are scarce, and the limited reports on the antagonistic relationship between the two species were addressed by in vitro experiments[37,38]. Previous studies have reported differential transcriptional control of matrix component expression in interactions between *Bacillus* and other bacterial species[38]. These findings support a hypothetical contribution of the extracellular matrix to the adaptation of *Bacillus* to the presence of other bacterial species, but no studies have directly demonstrated the functional significance of this bacterial structure in modulating such interactions.

In this work, we explore the functional role of the extracellular matrix of 3610 in the prevention of colony infiltration by PCL1606. Using time-lapse confocal microscopy we observe dramatic changes to cellular interactions between the two species in the absence of the *Bacillus* extracellular matrix. The combination of magnetic resonance imaging and solid-state nuclear magnetic resonance (NMR) reveals that the absence of an extracellular matrix leads to a less compact and more fluid colony, changes that may favor *P. chlororaphis* infiltration. Transcriptomic and metabolomics analysis identify the *Bacillus* lipopeptide surfactin and components of the *Pseudomonas* T6SS as additional candidates participating in this interaction. Analysis of plant co-colonization support the important role for the *Bacillus* extracellular matrix in determining bacterial distribution in mixed populations on leaves and a role for the *Pseudomonas* T6SS during plant seed germination.

## Results

**The extracellular matrix protects *Bacillus* from PCL1606 invasion.** To better understand the function of the extracellular matrix in bacterial interactions, we decided to study the interplay between 3610 and PCL1606. We initially evaluated the behavior of these strains with pairwise-interaction experiments using four different artificial media: King's B, a medium optimum for the growth and production of secondary metabolites of *Pseudomonas* strains; Msgg a medium optimum for the study of biofilm formation in *Bacillus*; and tryptone yeast (TY) or lysogeny broth (LB), two rich media routinely used for the growth of organotrophic bacterial species (Supplementary Fig. 1). Interactions after 72 h of growth showed the existence of a clear halo separating PCL1606 and 3610 colonies, an effect especially evident in King's B; and a reduction of wrinkles in the 3610 colony morphology. From this preliminary analysis, we decided to use LB medium to investigate the mechanism behind this bacterial interaction for two main reasons: first, the similar growth rate of both strains in LB (Supplementary Fig. 2), thus permitting a "balanced" interaction; and second, the apparent differences in *B. subtilis* biofilm morphologies.

We prepared single, double, and triple mutants of all the *B. subtilis* extracellular matrix components to investigate their respective contribution to the interaction with PCL1606. Pairwise time-course interactions between PCL1606 and WT 3610 revealed the existence of a subtle inhibition area between the two colonies, and a reduction of wrinkles in 3610 versus the strain growing alone (Fig. 1a, Supplementary Figs. 1d and 3a). Interactions with single mutants in *tasA* and the *tasA-bslA* double mutant were similar to those obtained for WT 3610 (Supplementary Fig. 4a, b, d). However, in single Eps mutants, and to a lesser extent for BslA, PCL1606 was able to penetrate the *B. subtilis* colony after 72 h (Supplementary Fig. 4c, f), and to partially colonize the frontline of the *Bacillus* colony after 96 h. This behavior was even more evident in the interaction between PCL1606 and the triple *eps, tasA, bslA* mutant (referred to as Δmatrix) where PCL1606 was able to completely colonize the *B. subtilis* colony after 96 h of interaction (Fig. 1b and Supplementary Fig. 3b). These findings strongly suggest a defensive role for the extracellular matrix, and that Eps and BslA are particularly important for this interaction.

**PCL1606 covers Δmatrix colonies without widespread cell death.** We had seen that the PCL1606-Δmatrix interaction culminates with PCL1606 penetrating the Δmatrix colony. We hypothesized that *B. subtilis* cells were being killed after physical contact between the colonies. To test this idea, we measured the colony-forming units (CFUs) for each species in three different areas of the interaction: a *Pseudomonas* area (PA) corresponding to the initial zone where the *Pseudomonas* colony was spotted (Fig. 2c and Supplementary Fig. 5c), an intermediate area (IA)

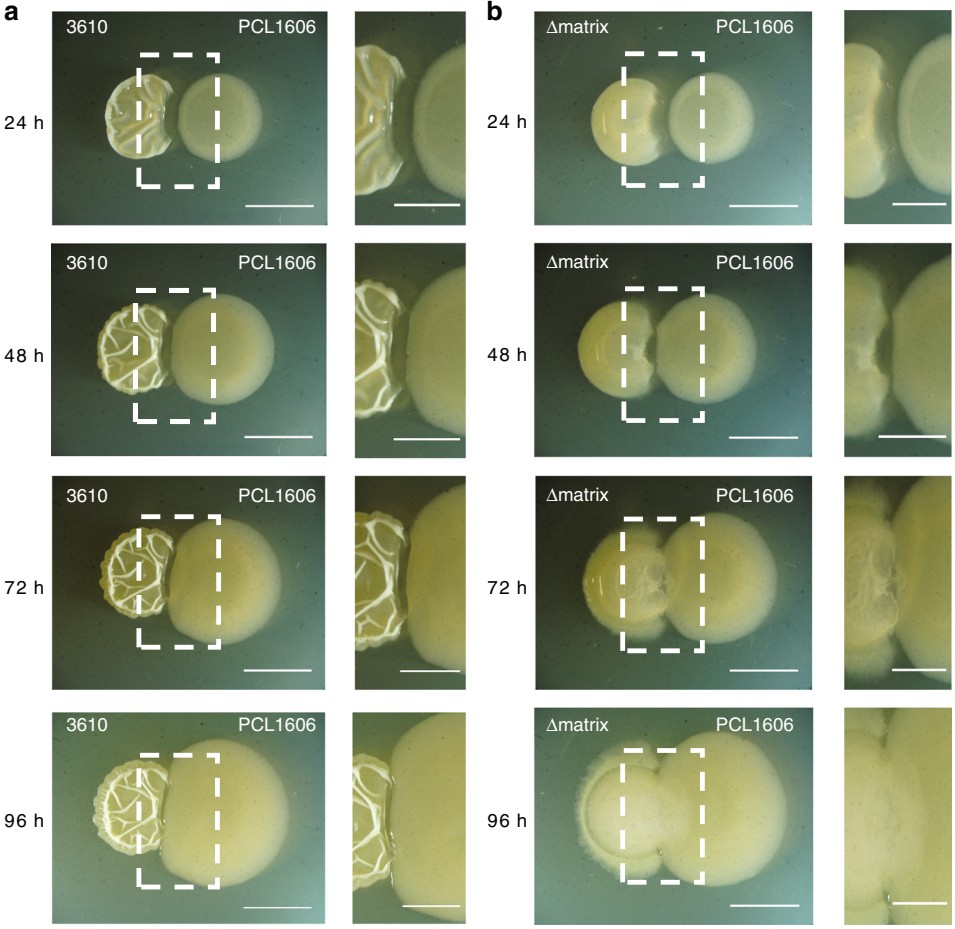

**Fig. 1** The lack of *B. subtilis* extracellular matrix permits PCL1606 overgrowth. **a, b** Time-courses of the interactions **a** 3610–PCL1606 and **b** Δmatrix–PCL1606 at 24, 48, 72, and 96 h growing in LB medium. Close-ups of the intermediate areas are shown close to each time-point of the interaction. Left colonies are *B. subtilis* strains and right colonies are PCL1606. Scale bar = 5 mm

corresponding to the zone where the first contact between the two colonies took place (Fig. 2b and Supplementary Fig. 5b), and a *Bacillus* area (BA) corresponding to the zone where the *Bacillus* colony was initially spotted (Fig. 2a and Supplementary Fig 5a). As expected from our macroscopic observations, no *B. subtilis* CFUs (either 3610 or Δmatrix) were obtained in the PA at any time during the interaction (Fig. 2c; and Supplementary Fig. 5a). The intermediate area was dominated by PCL1606 with percentages up to 95% of the total population (Fig. 2b and Supplementary Fig. 5b). Contrary to our expectations based on macroscopic observations, *Bacillus* CFUs were obtained at all time-points, and no differences in population size were evident between early (24 h) and later (96 h) stages of the experiment (Fig. 2b lower row and Supplementary Fig. 5b lower row). The most noticeable differences were observed in the BA. When we analyzed the pairwise-interaction between PCL1606 and 3610, *B. subtilis* CFUs made up virtually all the bacteria detected, with PCL1606 CFUs representing 0.007% of the total observed at later stages of the interaction (96 h), confirming that PCL1606 cannot easily penetrate 3610 colonies with a functional extracellular matrix (Supplementary Fig. 5a). As expected, in the Δmatrix–PCL1606 interaction, PCL1606 population was detected in the BA after 48 h, and progressively increased over time becoming the majority at 72 and 96 h (Fig. 2a). Surprisingly, the number of *Bacillus* CFUs remained unchanged during the time-course experiment, even after contact with PCL1606 (Fig. 2a lower row). One possibility is that invasion might trigger a survival mechanism such as sporulation in *Bacillus* bacteria[39].

Indeed, the sporulation rates of 3610 and Δmatrix bacteria increased dramatically from 58% and 30%, respectively, in monocultures (Supplementary Fig. 6a, b) to 95% in 3610 wild-type strain and 65% in Δmatrix, upon initial contact with PCL1606 after 48 h in the Intermediate Area (Fig. 2b, Supplementary Figs. 5b and 6a, b) and in the case of Δmatrix, reaching 95% in the BA after 72 h.

The fact that *B. subtilis* sporulation was triggered regardless of whether or not colonies were invaded by PCL1606 suggests that nutrient starvation might be one of the environmental cues that trigger sporulation. To test this idea, we mutated histidine kinases KinA, KinB, KinC, and KinD in 3610 and measured sporulation rates in both single colonies and in interactions with PCL1606 (Supplementary Fig. 7). The loss of KinA and KinB from 3610 resulted in sporulation levels during interaction with PCL1606 close to those seen in Δ*kinA* and Δ*kinB* single colonies, suggesting nutrient starvation as a trigger of *B. subtilis* sporulation in this interaction[40]. Thus, our data show that *B. subtilis* activates sporulation even when a functional extracellular matrix prevents colony invasion by PCL1606.

**Δmatrix and PCL1606 expand faster during their interaction.** To better understand the infiltration of Δmatrix colonies by PCL1606 cells, we decided to study this interaction at the cellular level using fluorescently labeled strains and time-lapse confocal laser scanning microscopy (CLSM). A PCL1606 strain constitutively-expressing DsRed and cyan fluorescent protein

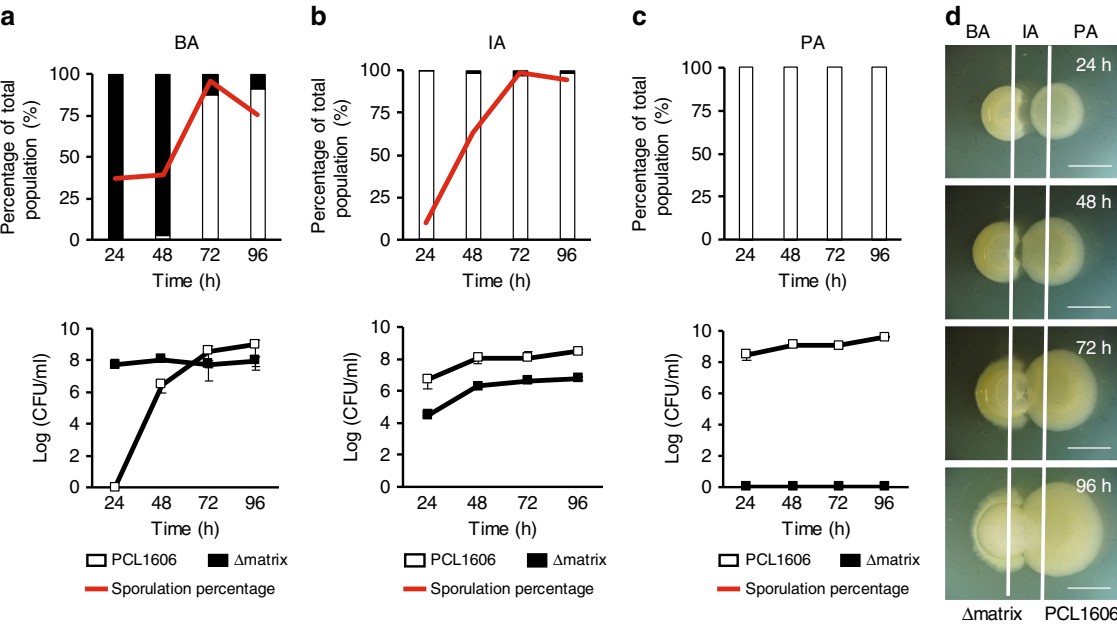

**Fig. 2** *B. subtilis* Δmatrix coexists with PCL1606 in the form of spores. **a–c** show (upper part) percentages and (bottom part) Log (CFU/ml) of PCL1606 (empty bars and squares) and Δmatrix (black bars and squares) in three different sections (BA [Δmatrix area], IA [Intermediate area], and PA [PCL1606 area]) of the interaction at different time-points (24 h, 48 h, 72 h, 96 h). Sporulation rates are shown as red lines. **d** Scheme of the interactions and the sections: BA (Δmatrix area), IA (Intermediate area) and PA (PCL1606 area). Scale bar = 5 mm. Cells from the three sections of the interaction were collected, sonicated and completely resuspended, diluted and plated to determine colony-forming units (CFU), sporulation rates and percentages of each species over the total bacterial population in the interaction. Average values of three biological replicates are shown, with error bars representing SD. Source data are provided as a Source Data file

(CFP)-expressing *B. subtilis* strains were spotted at a distance of 0.7 cm from each other on LB agar medium and their growth and interactions followed by time-lapse microscopy. Images were captured every 30 min over a total period of 4 days at multiple positions on the plate for each time-point. This procedure allowed us to define different stages of the interaction: independent colony growth, indirect interactions, first direct contact, and colony interaction resolution (Fig. 3). Furthermore, analysis of each time-lapse video allowed colony expansion rates to be calculated for each strain over the course of the interaction process.

Microscopic observations of the interaction between wild-type 3610 and PCL1606 were in line with our expectations based on the macroscopic analysis described above. During the first hours after being spotted onto the culture medium, wild-type 3610 cells started growing with a spatial distribution similar to the previously described Van Gogh bundles[41], with phases of advancement followed by phases of filling the colonized area (Fig. 3a and Supplementary Movie 1). During this period (the first 18–20 h), PCL1606 expanded, on average, faster than 3610 (0.013 μm/s vs 0.010 μm/s, respectively) (Fig. 3e–I). As the colonies approached, 3610 stopped expanding while PCL1606 continued spreading, however, the speed of the expansion progressively decreased until the colonies finally contacted after 30–35 h of growth (Fig. 3a, c, and e–II). From this time, a subtle and slow growth of PCL1606 over 3610 in the interaction zone was observed (Fig. 3a, c and e–III). Thus, under normal circumstances the two colonies retain their integrity and general organization, with a clearly defined boundary that changes only gradually.

Time-lapse imaging of Δmatrix and PCL1606 interactions uncovered a different interaction, providing some remarkable and unexpected insights into the process (Fig. 3b and Supplementary Movie 2). First, Δmatrix cells expanded significantly faster (0.017 μm/s) than the wild-type 3610 and PCL1606 (0.013 μm/s) strains during the first 10–16 h of the experiment (Fig. 3f–I), which

might indicate higher liquidity compared with wild-type 3610 colonies. Second, at the moment of first direct contact at 20–24 h, the speed of Δmatrix expansion unexpectedly increased and started to grow over the PCL1606 colony surface, an interaction that is macroscopically imperceptible (Fig. 3b, d). This critical stage can also be described by measuring the expansion rate of PCL1606, with negative values indicating a regression in the position of this strain on the solid medium (Fig. 3f–II). Third, when the interaction seemed to have stabilized after a few hours, CFP-labeled Δmatrix cells started disappearing from the interaction zone, an effect linked to sporulation of the *Bacillus* population (Fig. 3b and Supplementary Fig. 8). In parallel, we observed a concomitant emergence of PCL1606 spot-colonies, which resolved with complete colonization of the Δmatrix colony (Fig. 3b, d). Owing to the almost imperceptible nature of the early stages of invasion, presumably by individual *Pseudomonas* cells, it was not possible to accurately calculate speed measurements at this stage (Fig. 3f–III). Thus, the loss of extracellular matrix results in increased colony expansion speed but seemingly at the cost of reduced resistance to invasive bacteria.

**Loss of *Bacillus* Δmatrix integrity facilitates PCL1606 invasion.** We have shown that the extracellular matrix is necessary for protecting *Bacillus* from PCL1606 invasion and affects colony expansion dynamics. A simple explanation for both observations would be that the lack of an extracellular matrix provokes changes to the biophysical properties of the *Bacillus* colony that compromise its integrity and resistance to invasion. To test this idea, we performed magnetic resonance imaging (MRI) and solid-state NMR spectroscopy experiments comparing 3610 and Δmatrix colonies. MRI generates images based on the relaxation times of different areas following electromagnetic stimulation and provides a measurement of different properties such as water

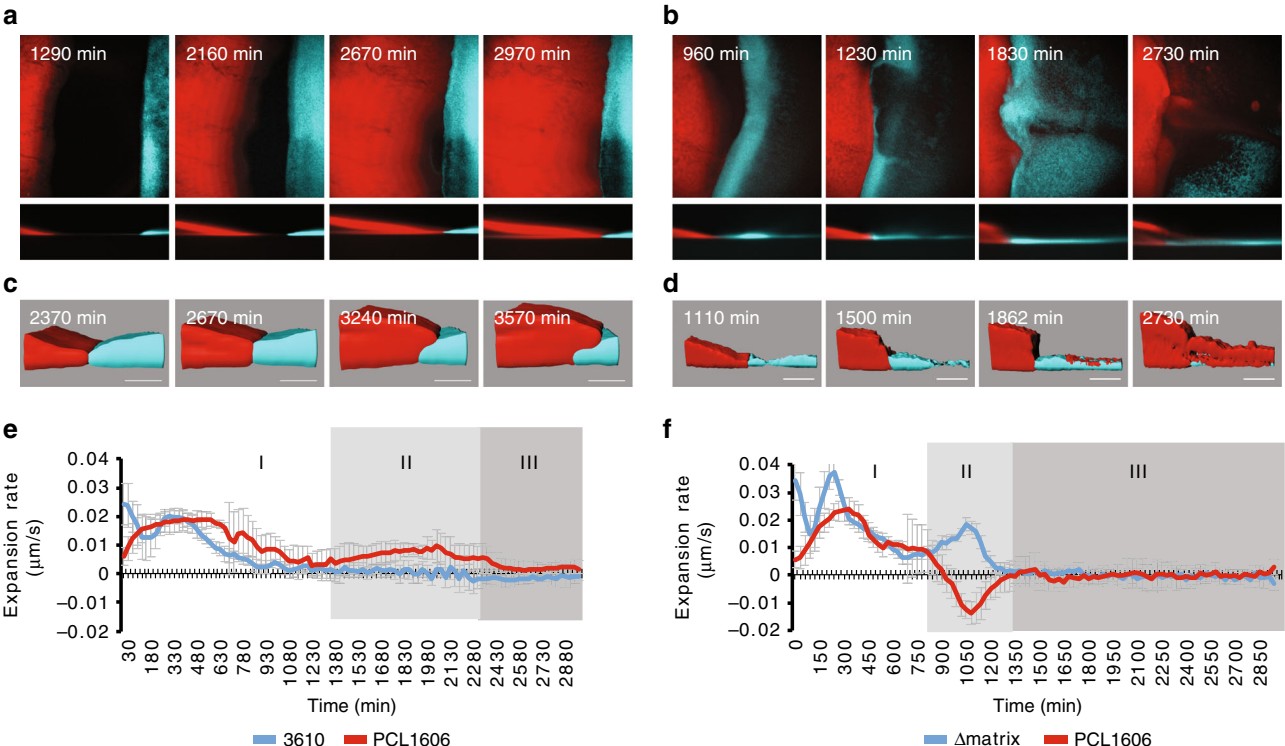

**Fig. 3** Colony expansion rates vary during the interaction. Red lines represent PCL1606 and blue lines indicate *Bacillus* strains. **a–d** CLSM time-course experiments of the interactions between **a**, **c** PCL1606 and 3610 and **b**, **d** PCL1606 with Δmatrix. **a**, **b** show the main steps during the interactions as a sum of the projections and in *z* axis. **c**, **d** show the most important points since both bacteria have interacted. **e**, **f** Expansion rates of the *Bacillus* and PCL1606 strains during the interactions can be divided in three different stages (I, II, III). Red lines represent PCL1606 and blue lines indicate *Bacillus* strains. **e** shows the expansion rates of 3610 and PCL1606 during the interaction. **f** shows the expansion rates of Δmatrix and PCL1606 during the interaction. Average values of four biological replicates are shown, with error bars representing SD. Scale bar = 100 μm. Source data are provided as a Source Data file

content and water diffusion in different colony regions. The high-resolution images obtained show that 3610 colonies exhibit high degrees of heterogeneity, with different areas in the same colony corresponding to the typical wrinkles found in the 3610 biofilm. In contrast, Δmatrix mutants did not exhibit any subzones, with water content being homogeneously distributed throughout the colony (Fig. 4a). $T_2$ relaxation times, which are inversely proportional to density of the medium, were analyzed in both colonies and higher values were obtained in Δmatrix (45 ms), whereas in 3610 the $T_2$ relaxation times varied depending on the area analyzed, with higher relaxation times in the wrinkles (43 ms) and the lowest values found at the edge of the colony (20 ms). In addition, we also measured the apparent diffusion coefficient (ADC), a value that is indicative of water diffusion in the colony and is inversely proportional to water movement restriction (Fig. 4b). Therefore, higher ADC levels are indicative of freer water movement and lower cellular density. Our results showed higher ADC levels in the Δmatrix colony than in 3610 ($2.3 \times 10^{-3}$ mm$^2$/s vs $1.5 \times 10^{-3}$ mm$^2$/s, respectively), confirming that the Δmatrix colony is less compacted than 3610, something that might favor penetration by PCL1606.

Complementary to the results obtained by MRI, we used solid-state NMR (SSNMR) to investigate the water-accessibility of 3610 and Δmatrix colonies. 1D cross-polarization $^{13}$C spectra (Fig. 4c) reveal well-resolved $^{13}$C signals for exopolysaccharides (e.g., glucose, *N*-acetyl-glucosamine, and galactose[42]), protein signals and lipids. Analysis of Δmatrix showed a decrease in amino-acid signals assigned to structural proteins TasA and BslA (signals at 175 and ~40 ppm and in the Cα and CH$_2$ aliphatic regions), and an overall decrease in the carbohydrate signals of the extracellular

matrix. In order to study polysaccharide hydration and water-accessibility in 3610 and Δmatrix colonies, we performed water-polysaccharide $^1$H spin diffusion experiments[43,44]. Results obtained using a mixing time of 10 ms that corresponds to ~ 50% of the maximal build-up intensity led us to detect rigid polysaccharide signals in close contact to water (Fig. 4d). 3610 and Δmatrix $^{13}$C-detected $^1$H-diffusion spectra recorded with a $^1$H-$^1$H mixing time of 10 ms (Fig. 4d) indicate a significantly different behavior, with weak signals for 3610 but much more intense signals for Δmatrix, reflecting increased accessibility to bulk water. $^{13}$C signals observed in the water-transferred spectra of Δmatrix mainly encode for *N*-acetyl-glucosamine (GlcNAc), suggesting that they are the most surface-exposed polysaccharides in the Δmatrix strain (Fig. 4e). This is consistent with the fact that the GlcNAc synthesis is not controlled by the *eps* operon mutated in the Δmatrix sample[45]. Moreover, GlcNAc is mainly found in the peptidoglycan of *B. subtilis* so we might assume that the absence of exopolysaccharides in the Δmatrix sample leads to an increase in peptidoglycan water accessibility.

**Bacillus surfactin distribution suggests a role in PCL1606 spread.** The biophysical differences caused by the presence or absence of an extracellular matrix highlight the role of this structure as a physical barrier and point to the extracellular matrix as the main element involved in the interaction with PCL1606. However, the altered kinetics of Δmatrix and PCL1606 colony interactions suggest that the situation may be more complex than anticipated. For example, the acceleration of Δmatrix and PCL1606 cells observed by CLSM at the earlier

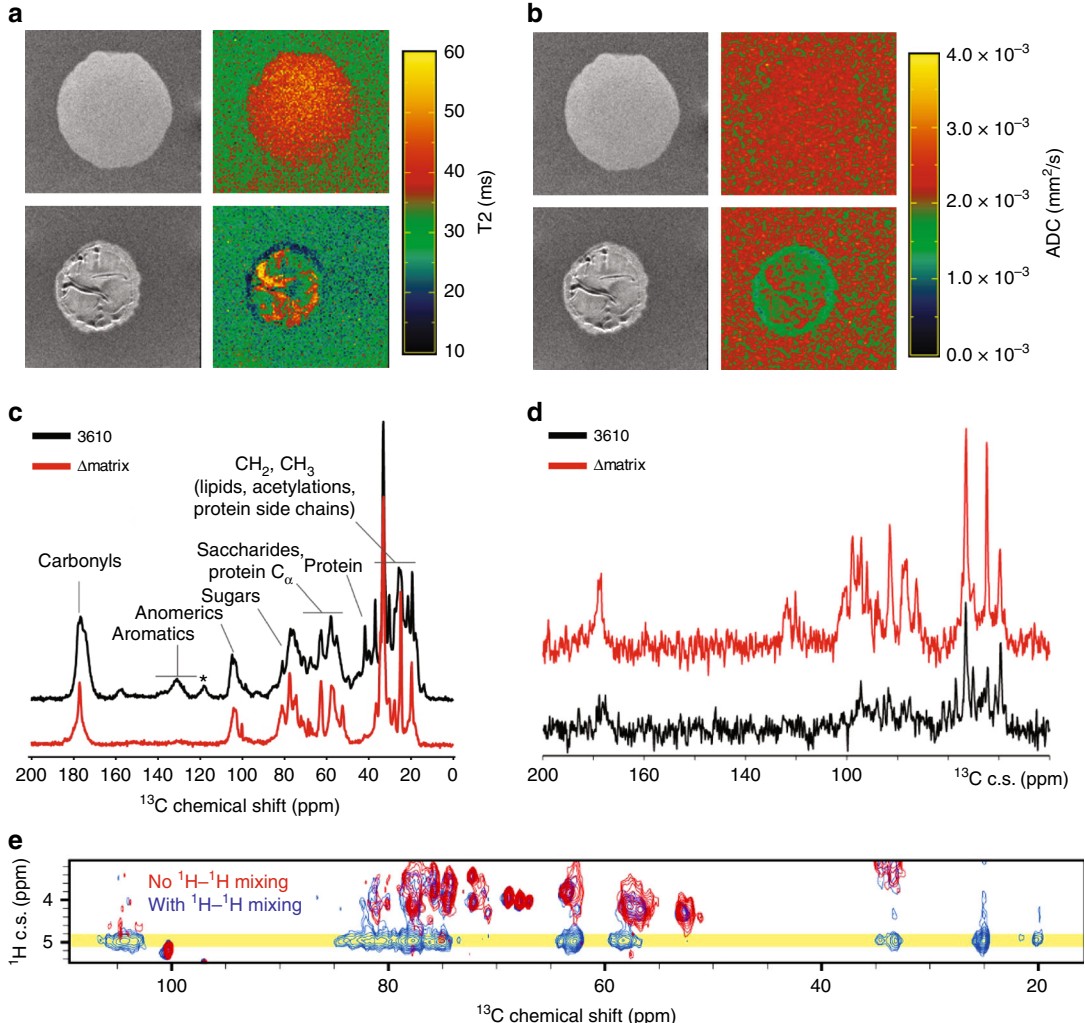

**Fig. 4** The *B. subtilis* extracellular matrix functions as a physical barrier. **a**, **b** Magnetic resonance imaging (MRI) analysis of Δmatrix and 3610 colonies after 72 h of growth on LB media. **c**–**e** Characterization by solid-state NMR of 3610 and Δmatrix grown on LB media after 72 h. **a** Transversal relaxation times ($T_2$) of Δmatrix and 3610 colonies. $T_2$-weighted (gray scale images) and $T_2$ map (heatmap) of Δmatrix and 3610 colonies. **b** Apparent diffusion coefficient (ADC) of Δmatrix and 3610 colonies. **c** $^{13}C$ cross-polarization spectra of $^{13}C$-labeled 3610 and Δmatrix samples. **d** $^{13}C$-detected spectra of the 3610 and the Δmatrix samples after a T2-filter and a 1H spin diffusion of 5 ms. **e** Superimposition of 2D $^{1}H$-$^{13}C$ HETCOR experiments of Δmatrix recorded without (in red) and with (in blue) a 1H spin diffusion mixing time. The bulk water frequency is indicated in yellow

stages of their interaction might indicate the presence of molecules that favor bacterial cell movement. To gain insight into factors that might affect colony expansion rates, we studied the spatial distribution of metabolites by performing imaging mass spectrometry analyses of single colonies and interactions at 24, 48, and 72 h (Fig. 5a–d). We did not see significant changes to the composition or spatial distribution of *Pseudomonas* metabolites between single colonies and wild-type and Δmatrix interactions (Supplementary Figs. 9–13). In general, most metabolites produced by *B. subtilis* showed similar distribution patterns in both interactions and monocultures (Supplementary Fig. 13), except for surfactin, which is not produced by PCL1606 (Supplementary Fig. 14a). A similar composition of surfactin isoforms was observed in the interactions of 3610 or Δmatrix with PCL1606, with the C13, C14, and C15-surfactin isoforms being the most abundant (Supplementary Figs. 14b, c, and 15), although their distribution patterns were different. In individual wild-type 3610 colonies, surfactin mainly accumulated outside of the colony at 24 and 48 h before decreasing after 72 h (Fig. 5a). Significantly, in single Δmatrix colonies, the distribution of surfactin was altered with similar levels present both inside and outside the colony,

with the biggest differences at 24 and 48 h (Fig. 5b), although relative amount of surfactins was similar comparing 3610 and Δmatrix colonies (Supplementary Fig. 15).

When we analyzed the distribution of surfactins during the PCL1606-Δmatrix interaction, we surprisingly found that they mostly localized to the interaction area but were also present in the Δmatrix and PCL1606 colony zones at 24 and 48 h (Fig. 5d). In contrast, surfactin from wild-type 3610 colonies during their interaction with PCL1606 accumulated strongly around the outside of the colony and throughout the PCL1606 colony but was largely excluded from the *Bacillus* colony itself (Fig. 5c).

These findings could reflect a role for surfactin as a factor that promotes *Bacillus* population motility and thus contributing to the penetration of *Bacillus* Δmatrix cells at the frontline with PCL1606 at early stages of the interaction, as visualized by microscopy (Fig. 3b, d). At the same time, an intriguing possibility is that PCL1606 may hijack surfactin to accelerate its motility and further infiltrate *Bacillus* colonies deprived of an extracellular matrix. To test this hypothesis, we mutated the genes involved in surfactin production in a Δmatrix background (ΔmatrixΔsurf) and performed pairwise-interaction assays with

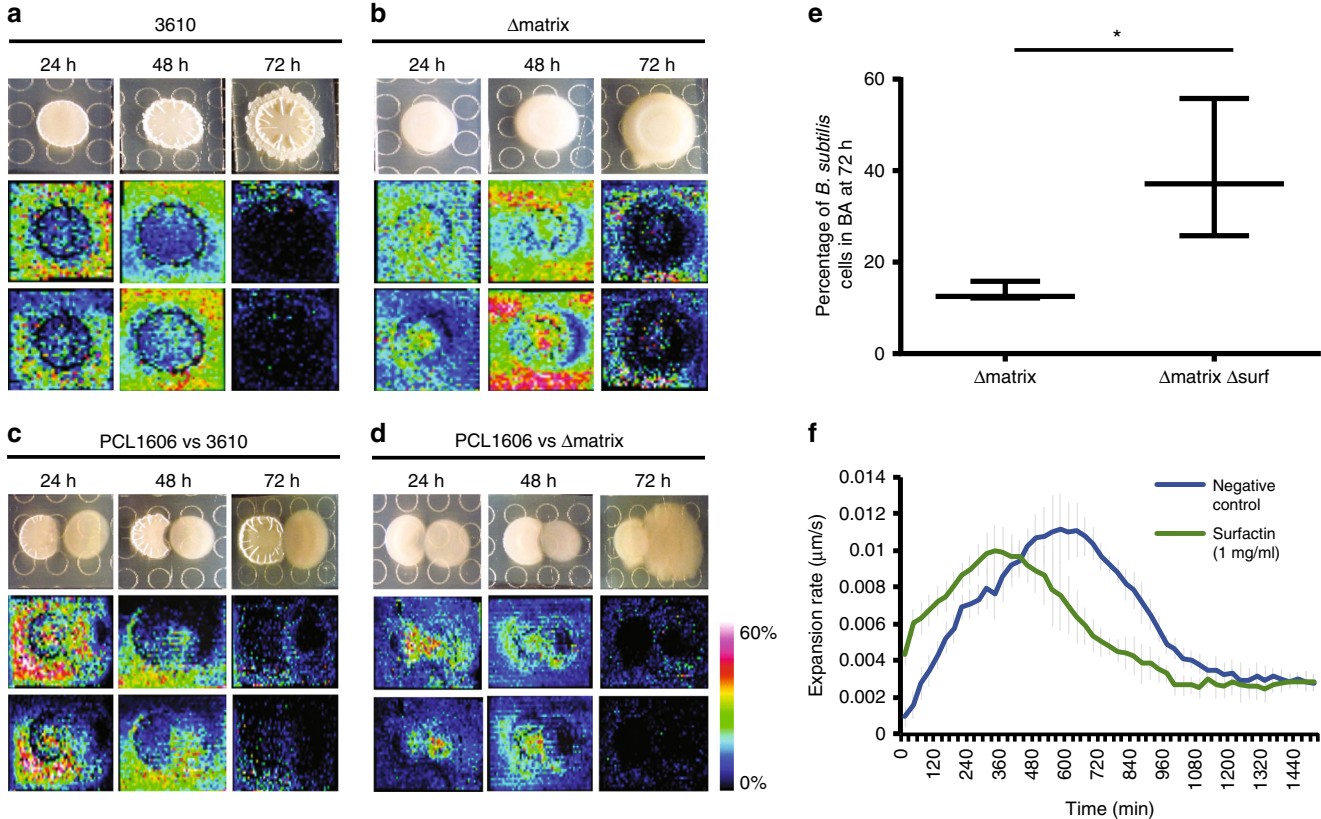

**Fig. 5** Spatial distribution of *Bacillus* surfactins accelerates the spread of PCL1606. **a–d** MALDI-TOF MSI time-lapse experiments (24 h, 48 h, 72 h) showed the spatial distribution of C14 and C15-surfactin isoforms (m/z 1058 and 1076 M + Na⁺) when growing **a** 3610 and **b** Δmatrix as single colonies or during the interactions between **c** PCL1606 (right colonies) and 3610 (left colonies) or **d** PCL1606 (right colonies) and Δmatrix (left colonies). **e** Comparison between the percentage of Δmatrix and ΔmatrixΔsurf populations in the *Bacillus* area (BA) after 72 h of interaction with PCL1606. *$P$ value < 0.05 ($t$ student). **f** Expansion rates of PCL1606 in the absence (blue line) or the presence (green line) of surfactin (1 mg/ml) spotted in a sterile disk ($n = 3$). Source data are provided as a Source Data file

PCL1606 (Fig. 5e). Results showed a clear reduction of PCL1606 cells in the BA after 72 h, suggesting a role for surfactin in facilitating PCL1606 penetration. Furthermore, the expansion rate of PCL1606 colonies in the interaction with ΔmatrixΔsurf and PCL1606 growing alone were similar during the first 10 h of interaction (0.014 μm/s and 0.014 μm/s, respectively), showing slower expansion rates than the PCL1606 colony in the interaction with Δmatrix (0.016 μm/s). In agreement with this result, we found that DsRed-labeled PCL1606 colonies expanded with altered kinetics when cultured in the presence of a disk impregnated with 1 mg/ml of surfactin compared to buffer (Fig. 5f). In the presence of surfactin, colony expansion was significantly faster during the earlier stages of the experiment. This effect appears to be relatively short term as control colonies were later able to expand at least as fast but only after a ~ 5 h delay. A possible explanation is that exogenous surfactin aids PCL1606 motility in the short term but that limiting factors, such as the rate of bacterial proliferation, may prevent a more sustained increase in the rate of colony expansion.

**PCL1606 activates the T6SS when in close contact with Δmatrix.** So far, our results suggest that *Pseudomonas* takes advantage of a deficient extracellular matrix and structural weakness together with the altered distribution of surfactin, to enter deeply into the colony, at which point most *Bacillus* undergo sporulation. Next, we wanted to use our in vitro model to determine, which mechanisms were being used by *Pseudomonas* to colonize Δmatrix colonies. We found that *Pseudomonas*

mutants unable to produce well-known secondary metabolites did not arrest the overgrowth phenotype (Supplementary Fig. 16). In an effort to identify other pathways and factors that might be involved in this process, we performed a dual RNA-seq analysis looking for changes in gene expression between individual colonies and the Δmatrix–PCL1606 interaction zone at 72 h. Mapping of the sequenced mRNAs to reference genomes showed that the vast majority of hits belonged to PCL1606, consistent with its larger CFU counts and the high sporulation rate of *Bacillus* in the interaction. Analysis revealed 318 differentially expressed genes in PCL1606 during the overgrowth (72 h of interaction), of which 62% were upregulated versus PCL1606 alone (Supplementary Fig. 17a).

The main transcriptional changes in PCL1606 can be categorized into three main groups: (i) amino acid and carbon metabolism, (ii) membrane composition, and (iii) the activation and repression of ABC transporters, efflux pumps, and signaling molecules (acyl-homoserine lactones, and siderophores such as pyochelin) (Fig. 6a, Supplementary Fig. 17b and Supplementary Data 1). Among the differentially expressed PCL1606 genes, we also found to be induced genes involved in the formation of the type VI secretion system (T6SS), which is known as an important mechanism in interactions and pathogenesis against bacterial and eukaryotic cells[27,28], but no studies have reported a role for this system against Gram-positive bacteria. Based on that, we decided to study the role of the T6SS in the interaction between *Pseudomonas* and *Bacillus* cells at a cellular level. To this end, the T6SS promoter was fused to the DsRed gene, introduced into PCL1606, and this strain tested in pairwise interactions against

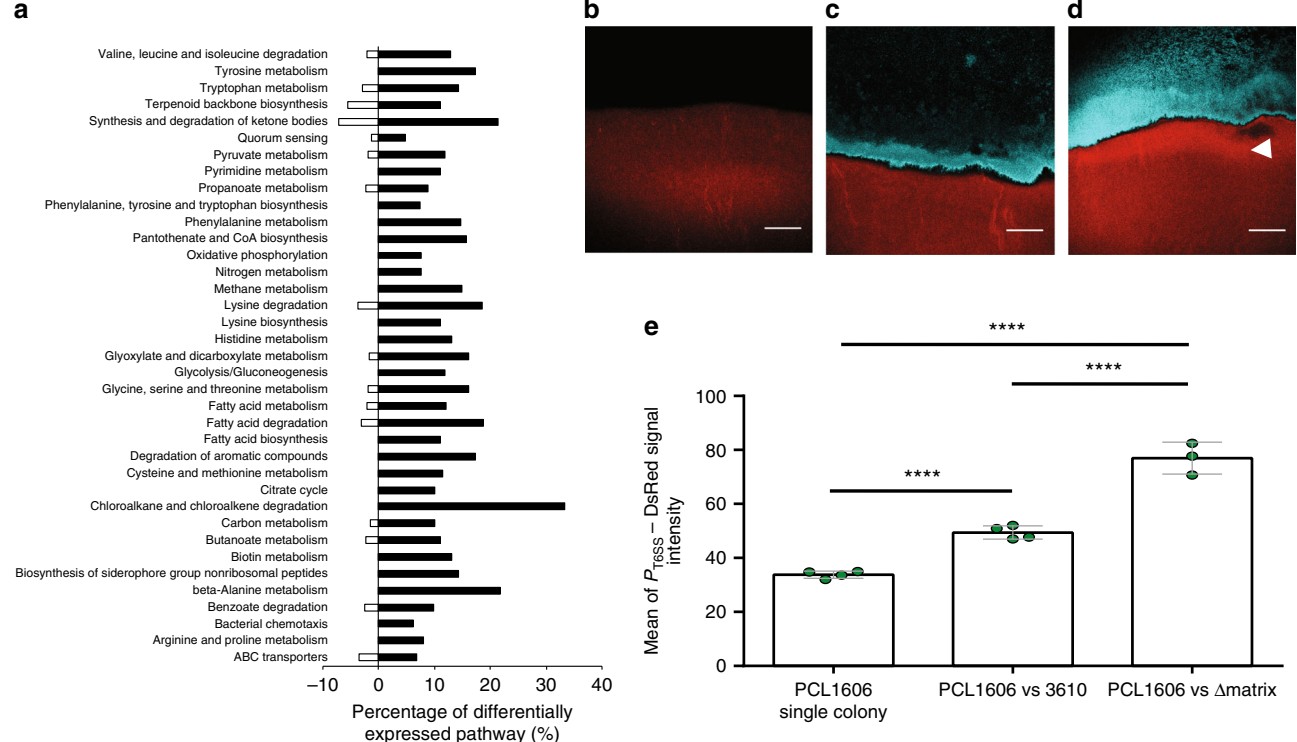

**Fig. 6** PCL1606 activates the T6SS upon contact with *B. subtilis* Δmatrix. **a** KEGG pathway analysis of the PCL1606 genes induced (black bars) and repressed (empty bars) after 72 h of interaction with Δmatrix. **b–d** Confocal scanning laser microscopy images of **b** the leading edge of PCL1606 $P_{T6SS}$-DsRed single colony, **c** intermediate area of the interaction between PCL1606 $P_{T6SS}$-DsRed colony and 3610 labeled with CFP, and **d** the interaction PCL1606 $P_{T6SS}$-DsRed vs Δmatrix labeled with CFP after 72 h of growth. White arrows highlight the areas where $P_{T6SS}$ is induced. Scale bar, 100 μm. **e** Measurement of the fluorescence intensity in the PCL1606 leading edge when growing alone or interacting with *Bacillus* strains. (*n* = 4). ****P value < 0.0001 (*t* student). Source data are provided as a Source Data file

CFP-labeled *Bacillus* strains. CLSM analysis revealed a high basal level of the $P_{T6SS}$ expression in the PCL1606 colony growing alone after 72 h (Fig. 6b, e). In agreement with the RNA-seq expression data, increased fluorescence signal accumulated in locations where PCL1606 and Δmatrix cells were in close contact (Fig. 6d, e). Interestingly, CLSM images of the interaction between PCL1606 and 3610 showed small $P_{T6SS}$ signal increase compared with the control one (Fig. 6c, e), whereas promoter expression was significantly higher in the interaction with Δmatrix.

To better determine the functionality of the T6SS for the invasiveness of *Pseudomonas*, we mutated the *tssA* gene (strain referred to as ΔT6SS), a key component of the T6SS baseplate and critical for the assembly and functionality of the T6SS[29]. Time-course interactions of Δmatrix and 3610 (Supplementary Fig. 18) in the interaction with ΔT6SS showed a pattern of penetration similar to that obtained for the wild-type PCL1606 strain, with infiltration and colonization of Δmatrix but not 3610 colonies. Furthermore, CFU counts were similar to those obtained in the interactions with the PCL1606 wild-type strain (Supplementary Figs. 5 and 19). However, sporulation rates were different in the interaction ΔT6SS vs Δmatrix (Fig. 7). Although induction of sporulation initially showed a pattern similar to the one observed with PCL1606, at later time-points of the interaction (96 h) we observed a decrease in the sporulated population of 47% and 30% in the Intermediate and BAs, respectively (Fig. 7a, b), whereas those levels remained at 95% when interacting with PCL1606 (Fig. 2b).

Altogether, these results highlight the importance of the T6SS in cell-to-cell contact with *B. subtilis* in the absence of extracellular matrix and in the alteration of *B. subtilis*

sporulation rates in the absence of a functional T6SS in PCL1606.

**Vegetative Δmatrix cells stop growing upon contact with PCL1606.** Our previous observations showed that, after 72 h of interaction with PCL1606, ~ 5% of the Δmatrix population would remain in the form of vegetative cells (Fig. 2a). Therefore, we expected the majority of the differentially expressed *Bacillus* genes in the RNA-seq would correspond to this 5%. This idea was supported by complementary observations: (i) at 72 h almost the entire population of Δmatrix had sporulated; (ii) microscopic observations revealed spores released from the mother cells (Supplementary Fig. 20), and (iii) the low efficiency of RNA purification from spores using standard protocols[46,47]. As described above for PCL1606, we compared gene expression between single Δmatrix colonies and *Bacillus* cells at the inter-action zone after 72 h, detecting 1105 differentially expressed genes of which 43% were upregulated (Supplementary Fig. 16a).

In contrast to the offensive strategy employed by *Pseudomonas*, the 5% of the *Bacillus* population that remained in a vegetative state arrested the growth, energy consumption and secondary metabolism together with the downregulation of sporulation and biofilm pathways where the Spo0A master regulator is involved (Supplementary Figs. 21 and 22). On the other hand, Δmatrix cells in the Intermediate Area activated the machinery of synthesis and reparation of DNA, the PBSX and SPβ prophages, chemotaxis and flagellar assembly, sulfur uptake and nitrogen metabolism, and competence genes such as *comG* and *comE* among other functions (Supplementary Data 2, Supplementary Figs. 21 and 22). This transcriptomic response appears indicative

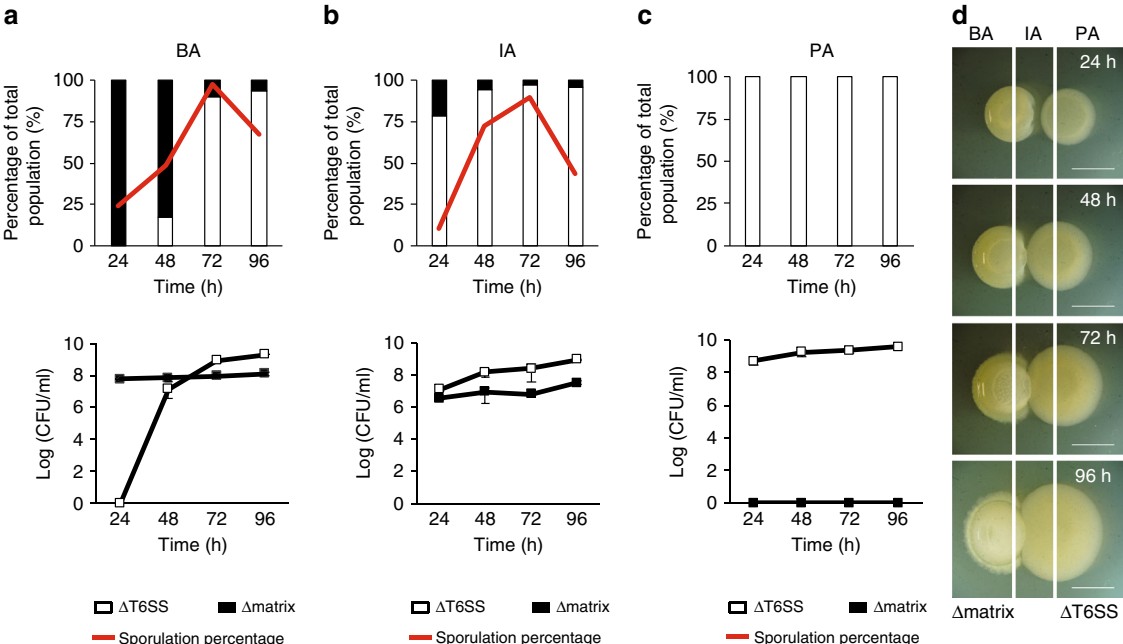

**Fig. 7** T6SS affects *B. subtilis* sporulation. **a–c** panels show (upper part) percentages and (bottom part) Log (CFU/ml) of ΔT6SS (empty bars and squares) and Δmatrix (black bars and squares) in three different sections (BA [Δmatrix area], IA [Intermediate area], and PA [ΔT6SS area]) of the interaction at different time-points (24 h, 48 h, 72 h, 96 h). Sporulation rates are shown as red lines. **d** Scheme of the interactions and the sections: BA (Δmatrix area), IA (Intermediate area), and PA (PCL1606 area). Scale bar = 5 mm. Cells from the three sections of the interaction were collected, sonicated and completely resuspended, diluted and plated to determine colony-forming units (CFU), sporulation rates and percentages of each species over the total bacterial population in the interaction. Average values of three biological replicates are shown, with error bars representing SD. Source data are provided as a Source Data file

of non-sporulating cells surviving PCL1606 colonization by different strategies.

**Bacillus and PCL1606 interact and co-exist on plant organs.** Our in vitro experiments have shown that *Pseudomonas* uses the T6SS as a powerful offensive strategy to exclude or compete with competitors, and that *Bacillus* responds to this behavior with a well-structured extracellular matrix and sporulation as a second line of a defensive strategy. To better understand the ecological significance of these two strategies, we examined these interactions on two anatomically and chemically different melon plant organs: the leaves, where bacterial cells have to adapt to limited nutrients and space (Fig. 8), and the seeds, where germination gives rise to an emergence of nutrients and the primary root or radicle (the first part of a seedling to emerge from a seed) that is susceptible for colonization by nearby bacterial species (Fig. 9).

We examined the persistence (CFU/ml) and colonization patterns of the different strains by CLSM at three time-points during the 9 days after bacterial inoculation on melon leaves. The persistence patterns of both *Bacillus* strains in melon leaves were similar in all the cases regardless of the single or co-inoculations performed, with small decreases in the bacterial populations after 2 or 9 days of inoculation (Fig. 8a). CLSM images of single inoculations showed similar distribution patterns in the three strains analyzed (Supplementary Fig. 23), whereas co-inoculations of PCL1606 with 3610 showed co-localization of the two species but with a stratified arrangement mostly in the intercellular zones of surface plant cells (Fig. 8c). Co-inoculation of Δmatrix and PCL1606 led to a more dispersed and heterogeneous distribution of cells lacking the horizontal stratification seen with 3610 (Fig. 8d). Quantification of bacterial distribution relative to leaf surfaces confirmed this change in organization with the Δmatrix population (Supplementary

Fig. 24b, d) being displaced further from the leaf surface versus the wild-type strain (Supplementary Fig. 24a, c) in agreement with a role for the *Bacillus* extracellular matrix as a physical barrier or support.

The higher levels of sporulation of both *Bacillus* strains in co-inoculated leaves compared with mono-inoculated ones suggests that the sporulation survival strategy observed in vitro is also active in more complex real-world environments such as leaves (Fig. 8b). However, in contrast to our in vitro observations, when assays were performed using ΔT6SS instead of the WT PCL1606 strain, a noticeable delay in the *Bacillus* sporulation rate was observed, although sporulation reached similar levels to those obtained in mono-inoculated leaves after 9 days, suggesting the involvement of the T6SS in the triggering of the *Bacillus* sporulation in plant leaves (Fig. 8b).

The same co-inoculation experiments were done in melon seeds, as both strains live in association with plant rhizospheres[34,48]. The CFU counts after 5 days of seed germination revealed a reduction of two orders of magnitude of the 3610 population when co-cultured with PCL1606 (Fig. 9a). Furthermore, as previously seen in melon leaves, the 3610 population had almost completely sporulated after 5 days alone or co-inoculated with PCL1606. However, the Δmatrix population showed remarkable sensitivity to the presence of PCL1606, with a sporulation rate rising from 27% in the mono-bacterized seeds to close to 100% in the co-bacterized samples (Fig. 9b). Interestingly, the Δmatrix population increased in one order of magnitude when co-inoculated with PCL1606 ΔT6SS compared to the single inoculated (Fig. 9a). In addition, the co-bacterization of 3610 or Δmatrix with the ΔT6SS resulted in the reduction of the sporulation rate, becoming almost negligible in Δmatrix (Fig. 9b).

These observations demonstrate that bacterial interactions are mediated by common mechanisms in vastly different

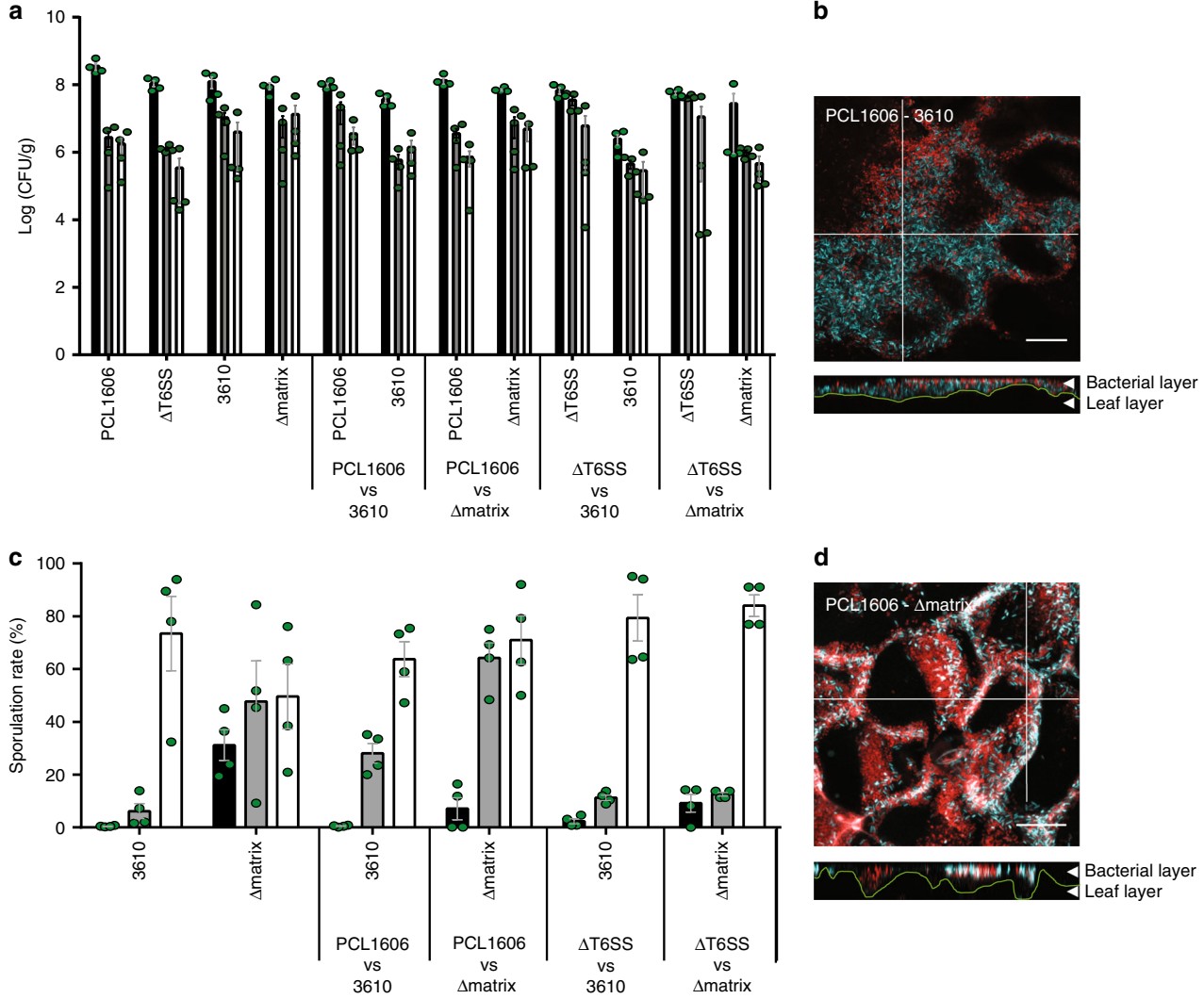

**Fig. 8** The *B. subtilis* ECM facilitates interaction with PCL1606 on leaves. **a** Log (CFU/ml) of *B. subtilis* (3610 and Δmatrix) and *P. chlororaphis* (PCL1606 and ΔT6SS) strains after single- and co-inoculation of melon leaves. **b** Sporulation percentages of *Bacillus subtilis* species after single- and co-inoculation of melon leaves. Colony-forming units and spore percentages were counted at 0 dpi (black bars), 2 dpi (gray bars), and 9 dpi (empty bars). Average values of four biological replicates are shown, with error bars representing SD. *P value < 0.05, **P value < 0.01, ***P value < 0.001 (Tukey test). **c, d** CLSM maximum projections and *z* axis slices at positions indicated by the white lines of the interactions between **c** 3610–PCL1606 (top) and **d** Δmatrix–PCL1606 (bottom) after 9 days of co-inoculation. *Bacillus* strains were labeled with CFP and PCL1606 was labeled with DsRed. Bacterial and leaf layers are indicated with arrows. Green lines indicate the leaf surface. White color is the consequence of the overlap between the fluorescence of PCL1606 and *Bacillus* strains. Scale bar, 100 μm. Source data are provided as a Source Data file

environments. Although the relative importance of these mechanisms may vary in different environments, we have shown that the *Bacillus* extracellular matrix and the *Pseudomonas* T6SS have important roles in establishing the balance between different bacterial populations. Further studies into the roles of other factors identified in this study, such as surfactin, may help uncover other ways in which bacteria defend themselves and even cooperate with other bacterial species.

## Discussion

Understanding the behavior and mechanisms involved in the interaction between bacterial species that share ecological niches is fundamental in the study of microbial population evolution and has important implications for biotechnology[49]. In their interaction with plants, it is important to decipher how intrinsically diverse microbes interacting with each other, the potential strategies used in host adaptation and how these strategies might co-

exist to produce a microbial consortium that benefits the host plant[50]. Factors that affect microbial interactions can be mediated via cell-to-cell contact or by diffusible molecules. Therefore, cellular spatial distribution is a key environmental factor in determining the result of these microbial interactions[51]. This suggests an important role for biofilms, which provide protection against harsh environments, enhance tolerance to physical and chemical stresses, and promote metabolic cooperation and the community-coordinated adjustment of gene expression[52,53]. In this work, we provide unprecedented insights into the fundamentals that underlie social interspecies interactions by studying how *B. subtilis* NCIB3610 and *P. chlororaphis* PCL1606 interact and co-exist in vitro and on plants.

In vitro experiments in optimal media for each bacterial species resulted in the physical exclusion of each species, and most importantly, the reduction of wrinkles, the most noticeable morphological feature of *B. subtilis* biofilms (Supplementary Fig. 1b, d). Based on our examination of pairwise interactions we

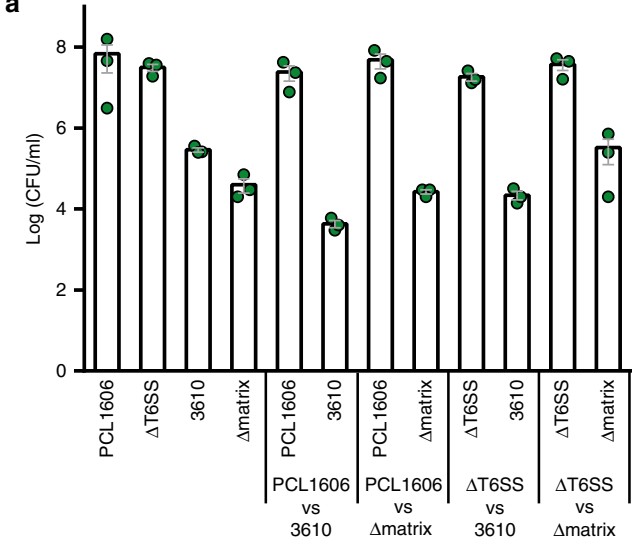

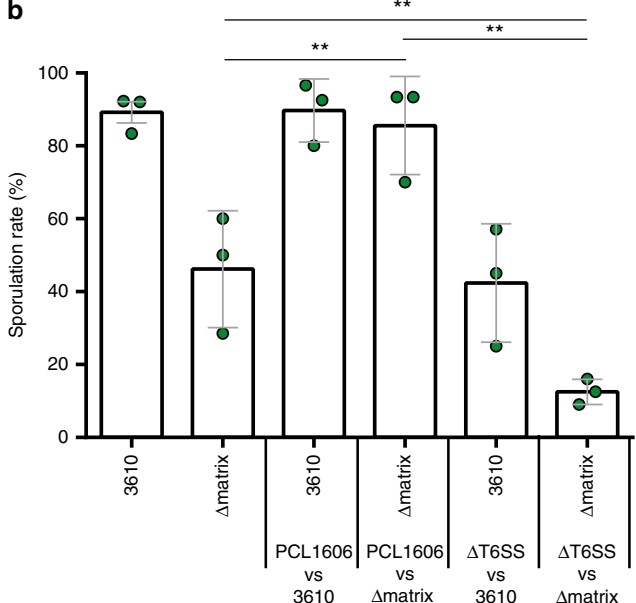

**Fig. 9** The T6SS affects *Bacillus* growth and sporulation in melon seed radicles. **a** Log (CFU/ml) of *B. subtilis* (3610 and Δmatrix) and *P. chlororaphis* (PCL1606 and ΔT6SS) strains after single- and co-bacterization of melon seeds. **b** Sporulation percentages of *Bacillus subtilis* strains after single- or co-inoculation with *Pseudomonas* strains of melon seeds. Colony-forming units and spore percentages of *Bacillus* were counted 5 dpi. Average values of four biological replicates are shown, with error bars representing SD. **P value < 0.01 (Tukey test). Source data are provided as a Source Data file.

have confirmed that the main components of the *Bacillus* extracellular matrix (Eps, TasA, and BslA) play a critical role in the protection of the *Bacillus* colony from the invasiveness of *Pseudomonas*. More specifically, BslA and exopolysaccharide contribute hydrophobicity and resistance to diffusible active molecules, respectively, to *B. subtilis* biofilms[16,20]. Thus, it is reasonable to think that the extracellular matrix might be physically impeding access of PCL1606 cells into the *Bacillus* colony, a notion supported by the overgrowth of PCL1606 on the Δmatrix colony (Fig. 1b). MRI and solid-state NMR data support the concept of the extracellular matrix as a physical barrier against PCL1606 as Δmatrix colony. In the absence of extracellular matrix MRI analysis shows a more fluid Δmatrix colony, a result that is in concordance with the increased water-accessibility

detected by solid-state NMR, an effect that could help PCL1606 penetrate the colony.

Examination of this interaction under the microscope allowed us to delineate more precisely the sequence of events at earlier stages, before macroscopic changes are visible, revealing details not observed in previous studies. First, the absence of extracellular matrix permitted faster expansion of the *Bacillus* colony, leading to the Δmatrix strain contacting PCL1606 earlier than the wild-type strain, and the partial penetration of *Pseudomonas* by the Δmatrix colony. Surfactin is a potent surfactant that reduces the water surface tension and promotes the social movement of *B. subtilis*[54,55]. Up to now, bacterial interactions involving surfactin-producing *Bacillus* species have suggested an inhibitory role for surfactin against other *Bacillus* species, or as an inhibitor of *Streptomyces* aerial hyphae development[56]. However, in this work we have presented data that suggests a role for surfactin in the interaction with PCL1606. Imaging mass spectrometry analysis suggests an accumulation of surfactin mostly in the zone of the initial contact and a distribution pattern altered in the Δmatrix strain. In contrast to the function described for interactions with other *Bacillus* species where surfactin showed an antimicrobial effect[57], in this case surfactin appears to collaterally promote PCL1606 colony spread. In fact, PCL1606 invasion was reduced when interacting with a non-surfactin-producing ΔmatrixΔsurf strain (Fig. 5e). Therefore, we propose that surfactin might act more generally as a mechanism to promote interspecies interaction, with further progression of the interaction, positive or negative, influenced by additional factors specifically employed by each bacterial species[36,58].

Further evaluation of the Δmatrix–PCL1606 interaction revealed the overgrowth phenotype of PCL1606 to be associated in part with its ability to metabolically adapt and interact with Δmatrix. An additional genetic change in PCL1606 upon interaction with *Bacillus* was activation of the T6SS. The T6SS encoded in the PCL1606 genome is genetically similar to the H2-T6SS of *Pseudomonas aeruginosa*[29,59–61] but with the peculiarity of having a PAAR gene and two genes of unknown functions in the middle of the T6SS cluster. Up to now, many works have shown a role for T6SS against Gram-negative bacteria species and eukaryotic cells[53–55], but no studies have reported the efficiency of the T6SS against Gram-positive bacteria. In addition, recent work has highlighted the relevance of the exopolysaccharide from *Vibrio cholerae* in the protection against external T6SS[62]. However, this is the first indication that T6SS-related factors influence sporulation in Gram-positive bacteria. Our findings indicate that the immunity of *Bacillus* to this offensive tool of PCL1606 seems to depend on the synthesis of an extracellular matrix, with the activation of sporulation as a secondary defense strategy in response to the PCL1606 T6SS.

Based on our studies, we propose two mechanisms that induce *Bacillus* sporulation depending on the distance between cells. The first mechanism is KinA and KinB-mediated, suggesting that *Bacillus* strains respond to nutrient availability and/or other environmental changes caused by the presence of PCL1606 (Supplementary Fig. 7), with no need for cellular contact. In addition, the involvement of molecules produced by PCL1606 in *Bacillus* sporulation, as shown in other cases with siderophores[63] and compounds such as decoynine and hadacidin[64], was also tested but purified pyochelin from PCL1606 failed to activate expression from the sporulation-related *sspB* promoter or change the rate of *B. subtilis* sporulation. Similar results were obtained with different fractions purified from PCL1606 cultures. Thus, results obtained suggest that the initial induction of *Bacillus* sporulation is most likely owing to environmental changes and nutrient availability during the interaction with PCL1606. The second strategy is most likely mediated by close contact between

*B. subtilis* and PCL1606 cells. In the scenario where *Bacillus* is deprived of the extracellular matrix and a cell-to-cell contact is occurring, the active presence of the T6SS plays a role in maintaining the high *Bacillus* sporulation levels as observed in in vitro and in planta experiments.

The results from the plant experiments support the value of our in vitro results, despite the obvious differences in the bacterial interactions observed. Our findings highlight the relevance of the *B. subtilis* extracellular matrix for surviving, persisting and colonizing two different plant organs[4], and the role of sporulation as a secondary defensive strategy when in close contact with PCL1606 cells. The T6SS of PCL1606 seems to influence *B. subtilis* sporulation in vitro (Figs. 2 and 7) and in vivo, delaying or reducing the sporulation rates of *Bacillus* strains in leaves and melon radicles when a PCL1606 ΔT6SS strain is used (Figs. 8b and 9b). In summary, our work increases our understanding of the mechanics of complex and multifactorial bacterial social interactions, demonstrating that the strategies adopted by two bacterial species may co-exist and lead to the formation of stable bacterial communities in plants, either mixed or as physically separated sub-domains. The findings obtained could be potentially relevant for agricultural purposes as shown by other studies[65,66] where the application of bacterial consortia has demonstrated promising effects in plant growth and biocontrol.

## Methods

**Strains, media, and culture conditions**. A complete list of the bacterial strains used in this study is shown in Supplementary Data 3. Routinely, bacterial cells were grown in liquid LB medium at 30 °C (PCL1606 and *B. subtilis*) or 37 °C (*E. coli*) with shaking on an orbital platform. When necessary, antibiotics were added to the media at appropriate concentrations. Strains and plasmids were constructed using standard methods[67]. Oligonucleotides used in this study are listed in Supplementary Data 4.

***Pseudomonas* T6SS mutant**. Chromosomal deletion of ImpA (TssA), a core T6SS baseplate component essential for its activity[68], was performed using the I-SceI methodology[69–71] in which upstream and downstream segments of homologous DNA are separately amplified and then joined to a previously digested pEMG vector using the Gibson Assembly Master Mix[72]. The resulting plasmid was then electroporated into PCL1606. After selection for positive clones, the pSEVA628S I-SceI expression plasmid was also electroporated. Co-integrated constructions were resolved by induction of I-SceI expression with 3 mM 3-methylbenzoate. Kanamycin-sensitive clones were PCR analyzed to verify the deletions. The pSEVA628S plasmid was cured by growth without selective pressure and its loss confirmed by sensitivity to 30 µg ml$^{-1}$ gentamicin and colony PCR screening as described by Martinez-Garcia and de Lorenzo[70].

***B. subtilis* mutants**. *B. subtilis* mutants were generated by SPP1 phage transduction as previously described[73]. To obtain the double mutant strains, phage lysates from *tasA* and *eps* single mutant strains were obtained and transferred to a *bslA* mutant. The same procedure was used to obtain the triple mutant, in this case using lysates from the *eps* single mutant and transferring it to a *bslA-tasA* double mutant. Mutants were confirmed both by PCR and by antibiotic resistance (kanamycin, MLS, and tetracycline in the case of the matrix mutant strain). Kinase mutants (*kinB*, *kinC*, and *kinD*) and *srfAA* mutant were obtained as indicated above for extracellular matrix mutants. The *kinA* mutant was obtained by phage transduction from *B. subtilis* JH12638 phage lysates[74].

**Construction of fluorescence labeling strains**. Fluorescence labeling plasmid pKM008V was constructed for *B. subtilis* strains. In brief, the P$_{veg}$ promoter fragment (300 bp) was extracted from pBS1C3 by digestion with *Eco*RI and *Hin*dIII restriction enzymes, purified, and cloned in pKM008 plasmid, which was previously digested with the same restriction enzymes. We used P$_{veg}$ as it is considered a constitutive promoter in *B. subtilis*. The same procedure was followed for mot promoter but the fragment was obtained by PCR using the 3610 chromosome as template.

pKM008V was then transformed into *B. subtilis* 168 by natural competence, and transformants selected by plating in LB plates supplemented with spectinomycin (100 µg ml$^{-1}$). Finally, the extracellular matrix mutant was fluorescently marked by transferring CFP from *B. subtilis* 168 using SPP1 phage transduction as previously described[73].

In the case of PCL1606 P$_{T6SS}$-DsRed, we fluorescently labeled the PCL1606 strain using pSEVA237D. In brief, we amplified the P$_{T6SS}$ promoter

region (250 bp) from PCL1606 genomic DNA. Plasmid and fragment were digested with *Eco*RI and *Hin*dIII restriction enzymes, purified, ligated, and cloned in *E. coli* DH5α competent cells. The completed pSEVA237D-P$_{T6SS}$ was then electroporated into PCL1606 cells. Introduction of the plasmid was confirmed by PCR and antibiotic selection (kanamycin).

**Pairwise interactions**. *B. subtilis* and *P. chlororaphis* strains were routinely spotted at a 0.7 cm distance onto an LB agar plate using 2 µl of cell suspension at an OD600 of 0.5. Plates were incubated at 30 °C and images taken at different time-points. Photographs were captured using a Leica M165 Stereomicroscope. For confocal microscopy time-course experiments, 0.7 µl of cell suspension were spotted at a 0.5 cm distance onto 1.2 mm thick LB agar plates using 35 mm glass bottomed dishes suitable for confocal microscopy (Ibidi).

**Bacterial population dynamics**. To analyze the cell-number percentage and growth curves during the interaction between *B. subtilis* (3610 and Δmatrix) and PCL1606 (PCL1606 and ΔT6SS), interaction plates were prepared as described, and incubated for the required time period. The interaction was then divided into three sections. The *B. subtilis* section (BA) consisted of the entire *B. subtilis* colony, not including the area proximal to the PCL1606 and ΔT6SS colonies. The interaction section (Intermediate Area) consisted of *B. subtilis* and PCL1606 and ΔT6SS cells more proximal to the other colony. The PCL1606 and ΔT6SS section (PA) consisted on the entire PCL1606 and ΔT6SS colonies excluding the cells included in the intermediate area. Each section was inserted into a 1.5 ml eppendorf tubes containing 1 ml phosphate-buffered saline (PBS), completely resuspended and subjected to mild sonication (three rounds of 20 sec pulses at 20% amplitude), serially diluted and plated onto LB petri dishes. Plates were incubated overnight at 30 °C. *Pseudomonas* and *Bacillus* colonies were easy to differentiate in terms of counts as they are morphologically different. For spores count, the serial dilutions previously mentioned were heated to 80 °C for 10 min in order to kill non-sporulated bacteria and plated as mentioned above.

**Whole-genome transcriptomic analysis**. Single colonies of PCL1606 and *B. subtilis* NCIB3610 (Δmatrix) were grown overnight in LB medium at 30 °C and spotted as single colonies or as interactions as previously described for 72 h. After that, cells were collected and stored at −80 °C. All the assays were performed in duplicate. Single colonies (control) were collected completely, while in pairwise interactions we only collected the area where *Bacillus* and *Pseudomonas* were mixed. For disruption of *Bacillus* single colonies and *Pseudomonas–Bacillus* mixed colonies, collected cells were resuspended in BirnBoim A[75] and lysozyme was added and incubated 15 min at 37 °C. After that, suspensions were centrifuged, the pellet resuspended in Trizol and total RNA extraction performed as indicated by the manufacturer. RNA extraction of *Pseudomonas* control colonies started in this point of the protocol as it is not necessary to add lysozyme for cell disruption. DNA removal was carried out by treatment with Nucleo-Spin RNA Plant (Macherey–Nagel). Integrity and quality of total RNA was assessed with an Agilent 2100 Bioanalyzer (Agilent Technologies). Removal of rRNAs was performed using RiboZero rRNA removal (bacteria) kit from Illumina and 100-bp single-end reads libraries were prepared using a TruSeq Stranded Total RNA Kit (Illumina). Libraries were sequenced using a NextSeq550 sequencer (Illumina).

Raw reads were quality-trimmed with cutadapt. Then, trimmed reads were mapped against *P. chlororaphis* PCL1606, *B. subtilis* 3610 reference genomes using EDGE-pro[65] software, which is specially designed to process prokaryotic RNA-Seq data. Quantification was also performed with EDGE-pro. Raw counts were normalized using the trimmed mean of M-values (TMM) method[76], implemented in NOISeq R package[77]. Differential expression analyses were performed with DESeq2 package[78]. Genes were considered as differentially expressed when logFC was higher than 1 and lower than −1, and p value < 0.05.

**Confocal laser scanning microscopy**. Bacterial interactions in solid medium were visualized by Confocal Laser Scanning Microscopy (CLSM). For the observation of *B. subtilis* and PCL1606 strains (labeled with CFP and DsRed, respectively) growing on melon leaves, 30 mm diameter discs were obtained with a cork-borer. A drop of glycerol was applied onto a labeled section each colony and placed onto 1.5 thickness cover glass (22 × 22 mm) and sealed with adhesive tape to a standard glass microscope slide. In both cases, images were obtained by visualizing samples using a Leica SP5 confocal microscope with a × 40 NA 1.3 Plan APO oil-immersion objective. Image processing was performed using Leica LAS AF (LCS Lite, Leica Microsystems) and FIJI/ImageJ software. For each experiment, laser settings, scan speed, photomultiplier detector gain, and pinhole aperture were kept constant for all acquired image stacks.

For bacterial interaction time-course experiments, *B. subtilis* and *P. chlororaphis* labeled strains were placed on a thin (1.2 mm) layer of LB agar in 35 mm glass bottom dishes (Ibidi) suitable for microscopy. Plates were incubated at 30 °C for 6 h prior to acquisition. Temperature was maintained at 30 °C during the time-course using the integrated microscope incubator. Acquisitions were performed using an inverted Leica SP5 confocal microscope with a × 25 NA 0.95 NA IR APO long working distance water immersion objective. Bacterial fluorescence could be visualized from underneath the bottom of the plate and through the agar medium

thanks to the long 2.2 mm free working distance of this objective. A special oil immersion medium, Immersol W 2010 (Zeiss) was used instead of water to avoid problems with evaporation during the experiment. Colony fluorescence was followed in multiple regions selected at the start of the experiment, with the acquisition of a series of different focal (z) positions at each region performed automatically at every time-point. Evaporation from the LB agar and its utilization by the growing colonies resulting a gradual lowering of the agar surface relative to the objective lens of ~ 250 microns every 24 h. In order to be able to follow colony dynamics, images were acquired over wide focal range to compensate for the predicted change in colony position during the experiment. Image processing and three-dimensional (3D) visualization was performed using ImageJ/FIJI[79,80] and Imaris version 7.6 (Bitplane).

For the interaction between 3610 or Δmatrix labeled with CFP and PCL1606 $P_{T6SS}$-DsRed and Δmatrix $P_{sspB}$-YFP versus PCL1606, experiments were performed as indicated in time-course experiments and images were taken at desired time-points. Colony growth speeds were calculated with FIJI, using the change in position of the leading edge of the colony between time-points to calculate its speed in microns per seconds. Results were calculated as an average speed at three different regions of the same colony, with variations expressed as standard deviation. To reduce the impact of random vibrations and variations, plotting speed values were smoothed as a floating four-value average advancing 30 min (or one time-point) at a time.

**MRI**. B. subtilis colonies (3610 and Δmatrix) were grown on LB media for 72 h. After that, colonies were covered with 1.5% agarose and subjected to analysis. All the MRI experiments were carried out on a 9.4 T Bruker Biospec system equipped with a 400 mT m$^{-1}$ gradients and an Avance III console (Bruker BioSpin, Ettlingen, Germany). High-resolution T2-weighted images were acquired using a turbo-RARE sequence (TE = 33 ms, TR = 500 ms, 2 averages, FOV = 3.2 cm, matrix size = 384 × 384, 78 mm in-plane resolution, and 1 mm slice thickness).

**Solid-state NMR spectroscopy**. B. subtilis colonies (3610 and Δmatrix) were grown on LB supplemented with $^{13}C$-labeled glucose for 72 h to provide a $^{13}C$-enrichment and were filled in 4 mm SSNMR rotors. Experiments were performed on a 600 MHz Bruker Biospin spectrometer equipped with a 4 mm Efree probe at a magic-angle spinning frequency of 11 kHz. Cross-polarization spectra were recorded using 1024 scans and a contact time of 1 ms. Water-edited $^{13}C$ spectra (256 scans) were recorded using a $T_2$-filter of 1.5 ms followed by a variable $^1H$ mixing time. The 2D $^1H$-$^{13}C$ HETCOR experiments were recorded with and without a 5 ms $^1H$-$^1H$ mixing time using 128 scans and acquisition times of 7.5 ms (indirect) and 20 ms (direct).

**Matrix-assisted laser desorption ionization imaging mass spectrometry (MALDI-TOF MSI)**. To perform MALDI-TOF MSI a small section of LB agar containing the cultured microorganisms (both in single colonies and in interactions) were cut and transferred to a MALDI MSP 96 anchor plate. Deposition of matrix (1:1 mixture of 2,5-dihydroxybenzoic acid and α-cyano-4-hydroxycinnamic acid) over the agar was done using a 53 μm molecular sieve. After that, plates were dried at 37 °C for 4 h. Photographs were taken before and after matrix deposition. Samples were analyzed using a Bruker Microflex MALDI-TOF mass spectrometer (Bruker Daltonics, Billerica, MA, USA) in positive reflectron mode, with 300–400 μm laser intervals in X and Y directions, and a mass range of 100–3200 Da. Data obtained were analyzed using FlexImaging 3.0 software (Bruker Daltonics, Billerica, MA, USA). The acquired spectra were normalized by dividing all the spectra by the mean of all data points (TIC normalization method). The resulting mass spectrum was filtered manually in 0.25% (3.0 Da) increments assigning colors to the selected ions associated to the metabolites of interest.

**Liquid chromatography–tandem mass spectrometry (LC-MS/MS)**. Non-targeted LC-MS/MS analysis was performed on a Q-Exactive Quadrupole-Orbitrap mass spectrometer coupled to Vanquish ultra-high performance liquid chromatography (UHPLC) system (Thermo Fisher Scientific, Bremen, Germany) according to Petras, Nothias[81]. Therefore, 5 LC-MSL of the samples were injected on UHPLC separation on C18 core-shell column (Kinetex, 50 × 2 mm, 1.8 μm particle size, 100 Å pore size, Phenomenex, Torrance, USA). For the mobile phase we used a flow rate of 0.5 mL/min (Solvent A: H2O + 0.1% formic acid (FA), Solvent B: Acetonitrile + 0.1% FA). During the chromatographic separation, we applied a linear gradient from 0–0.5 min, 5% B, 0.5–4 min 5–50% B, 4–5 min 50–99% B, flowed by a 2 min washout phase at 99% B and a 2 min re-equilibration phase at 5% B. For positive mode MS/MS acquisition the electrospray ionization was set to a 35 L/min sheath gas flow, 10 L/min auxiliary gas flow, 2 L/min sweep gas flow and 400 °C auxiliary gas temperature. The spray voltage was set to 3.5 kV with an inlet capillary of 250 °C. The S-lens voltage was set to 50 V. MS/MS product ion spectra were acquired in data dependent acquisition (DDA) mode. MS1 survey scans (150–1500 m/z) and up to five MS/MS scans per DDA duty cycle were measured with a resolution (R) of 17,500. The C-trap fill time was set to a maximum of 100 ms or till the AGC target of 5E5 iones was reached. The quadrupole precursor selection width was set to 1 m/z. Normalized collision energy was applied stepwise at 20, 30, and 40% with z = 1 as default charge state. MS/MS scans were triggered with apex mode within 2–15 s from their first occurrence in a survey

scan. Dynamic precursor exclusion was set to 5 s. Precursor ions with unassigned charge states and isotope peaks were excluded from MS/MS acquisition.

**Data analysis and MS/MS network analysis**. After LC-MS/MS acquisition, raw spectra were converted to.mzXML files using MSconvert (ProteoWizard). MS1 and MS/MS feature extraction was performed with Mzmine2.30[82]. For MS1 spectra, an intensity threshold of 1E5 and for MS/MS spectra of 1E3 was used. For MS1 chromatogram building a 10 ppm mass accuracy and a minimum peak intensity of 5E5 was set. Extracted Ion Chromatograms (XICs) were deconvoluted using the baseline cutoff algorithm at an intensity of 1E5. After chromatographic deconvolutiuon, XICs were matched to MS/MS spectra within 0.02 m/z and 0.2 min retention time windows. Isotope peaks were grouped and features from different samples were aligned with 10 ppm mass tolerance and 0.1 min retention time tolerance. MS1 features without MS2 features assigned were filtered out the resulting matrix as well as features, which did not contain isotope peaks and which did not occur at least in three samples. After filtering gaps in the feature matrix were filled with relaxed retention time tolerance at 0.2 min but also 10 ppm mass tolerance. Finally, the feature table was exported as.csv file and corresponding MS/MS spectra as.mgf file. Contaminate features observed in Blank samples were filtered and only those whose relative abundance ratio blank to average in the samples was lower than 50% were considered for further analysis.

For molecular networking and spectrum library matching the.mgf file was uploaded to GNPS (gnps.ucsd.edu)[83]. For molecular networking, the minimum cosine score was set as 0.7. The Precursor Ion Mass Tolerance was set to 0.01 Da and Fragment Ion Mass Tolerance to 0.01 Da, Minimum Matched Fragment Peaks was set to 4, Minimum Cluster Size to 1 (MS Cluster off) and Library Search Minimum Matched Fragment Peaks to 4. When Analog Search was performed the Cosine Score Threshold was 0.7 and Maximum Analog Search Mass Difference was 100. Molecular networks were visualized with Cytoscape version 3.4[84].

**Bacterial interactions on plant leaves**. The assay was set up as previously reported[85]. In brief, melon seeds of cv. Rochet were pre-germinated, sown into pots, and cultivated in a plant growth chamber until use. Before each experiment, bacterial cultures were incubated overnight at 28 °C in an orbital shaker, two-time washed, and cultures adjusted to a cell density of 10$^8$ cfu ml$^{-1}$. In the case of Bacillus and Pseudomonas co-inoculation, cultures were mixed prior to leaf application. Plants were then incubated in a growth chamber under controlled conditions (25 °C over a 16–8 h photoperiod). Leaves were collected at 4 h, 2 days, and 9 days post inoculation, fresh weight was measured and CFU and spore percentages calculated. 3D acquisitions of melon leaf surfaces were acquired with a × 40 1.30 NA Plan APO oil immersion objective and Leica SP5 confocal microscope. CFP-positive 3610 and DsRed-positive 1606 bacteria were detected automatically using the Imaris (version 7.6.5) spot detection algorithm using an estimated diameter of 1.26 μM and background subtraction. Identical CFP and DsRed intensity thresholds were used between samples. Leaf surfaces were manually defined as an Imaris surface object and used to calculate the number of CFP and DsRed-positive bacteria at distances of 1–9 μM from the leaf surface at 1-micron intervals using the Imaris "Find Spots Close to Surface" function. At all time-points, leaf discs were taken for confocal microscopy analysis. Experiments were repeated at least three times.

**Seed colonization assays**. Melon seeds were bacterized with mono- and co-cultures of B. subtilis and PCL1606 for 1 hour. Competitive colonization assays were performed as previously described[86], using single strains and 1:1 mixes of B. subtilis and PCL1606 strains. Seeds were grown for 5 days at 25 °C before bacterial persistence quantification and size, weight, and area calculation. For bacterial quantitative analysis, roots were cut, weighed, and introduced in eppendorfs with 1 ml of M9 basal medium and 1 g of glass beads (diameter 3 mm). Bacteria attached to the roots were recovered by vortexing for 1 min, and CFUs and sporulation percentages were calculated by plating serial dilutions of the resulting suspension on LB medium. Antibiotics supplementation was not needed as colonies were easily differentiated by morphology. Experiments were repeated at least three times.

**Statistical analysis**. Results are expressed as mean ± SD. Statistical significance was assessed using Tukey's or student t tests. All analyses were performed using GraphPad Prism® version 6. P values < 0.05 were considered significant.

**Reporting summary**. Further information on research design is available in the Nature Research Reporting Summary linked to this article.

## Data availability

All LC-MS/MS data were deposited to the Mass spectrometry Interactive Virtual Environment (MassIVE) at https://massive.ucsd.edu/ with the identifier MSV000082402. Molecular Networking and spectrum library matching results can be found online at the GNPS webpage under the following links: https://gnps.ucsd.edu/ProteoSAFe/status.jsp?task=705915a36bd24cce9dbcedd4b876b008 and https://gnps.ucsd.edu/ProteoSAFe/status.jsp?task=2c41ab80bb12490ba52fa7e8c08fd56b. RNA-seq data have been deposited in the GEO database: GSE117802. The source data underlying Figs. 2, 3, 5, 7, 8, and 9 and Supplementary Figs. 2, 5, 6, 7, 8, 19, and 24 are provided as a Source Data file.

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

## Acknowledgements

We thank the Ultrasequencing Unit of the SCBI-UMA for RNA sequencing, the Bioinformatic unit of GENYO (Granada, Spain) for the analytical treatment of the data. We are grateful to Marta Martínez-Gil (University of Málaga) and Paul Straight (Texas A&M University) for critical reading and multiple suggestions and comments on the manuscript, and Luis Díaz (University of Málaga) for technical support. We also thank Ákos T. Kovács (Technical University of Denmark) for kindly provide *B. subtilis* JH12638 strain. This work was partially supported by grants from ERC Starting Grant (BacBio 637971) and Plan Nacional de I + D + I of Ministerio de Economía y Competitividad (AGL2016-78662-R). C.M.S is funded by the program Juan de la Cierva Formación (FJCI-2015-23810). M.V.B.C. is funded by the program Plan Propio de Investigación y Transferencia from Universidad de Málaga. Furthermore, we thank Ministerio Economía y Competitividad for their support through grant AGL2014-5218-C2-1-R to F.M.C. and the US National Institutes of Health (NIH) for their support through grants GMS10RR029121, 5P41GM103484–07 to P.C.D. and the German Research Foundation (DFG) with Grant PE 2600/1 to D.P.

## Author contributions

D.R. conceived the study; D.R and C.M.S. designed the experiments; C.M.S. performed the main experimental work; C.M.S. and J.P. performed and designed the confocal microscopy work and data analysis; Y.N.G. and C.M.S. constructed *Bacillus* strains, C.M.S. and M.V.B.C. performed the experimental work related with seeds; C.M.S., D.P., P.C.D. and A.M.C.R. performed MALDI-TOF MSI and LC-MS/MS analysis; M.L.G.M. performed MRI experiments; A.L., B.H. and G.L. performed solid-state NMR experiments; D.R., J.P. and C.M.S. wrote the manuscript; and J.P., A.V., F.M.C. and P.C.D. contributed critically to writing the final version of the manuscript.

## Additional information

**Competing interests:** The authors declare no competing interests.

