## [Peer Review File · Nature Communications]

Reviewers' comments:

Reviewer #1 (Remarks to the Author):

The manuscript present an interesting observation: colonies of *Bacillus subtilis* and *Pseudomonas chlororaphis* show a defined interaction pattern which is altered if *B.subtilis* is lacking matrix production. In a series of in vitro LB plate based experiment, the authors demonstrate that deletion of the 3 matrix component has great impact on sporulation in the colonies, spatial distribution of surfactin, while inactivation of T6SS in *Pseudomonas* alters this interaction. Finally, they show that matrix deletion in *Bacillus* or T6SS deletion in *Pseudomonas* also alters the interaction and mostly the sporulation efficiency when inoculated on melon leaves or seeds. There are lots of very interesting information in the manuscript which all suggest how the two species interact.

There are few grammar mistakes in the manuscript. In general, the manuscript might require a professional text/grammar editor so the storyline can be easier followed.

Major comments:

All colony images are lacking scale bars and should be included in each figures with colony images.

There are some inconsistencies in the colony sizes among images. For example, fig 1d PCL colony is seems smaller at 96h compared to 72h, similarly fig 4c and d: PCL at 48 h seems smaller than 24 h samples. How consistent were the colony sizes? Is this due to the use of a normal camera instead of a zoom microscope, where magnification (and therefore also scalebar) can be controled and recorded.

Fig 2a-e seems to fully represent the figure panels in Supplemental Fig 8a-e - why is this the case? This suggest hardly any impact of T6SS mutation on the interaction with the Dmatrix strain

Is the T6SS activated when in contact with 3610 strain? Reporter measurement is missing for the combination of PCL and 3610. Also, does the interaction change between 3610 and *tssA* mutant PCL?

Generally, it is well established that matrix non-expressing cells show high motility (e.g. *ymdB* mutant lacking matrix production). Thus it is hardly surprising that motile cell clusters are detectable. I am missing a proper control to demonstrate that in the absence of PCL, motility is not expressed highly in the Dmatrix strain. The authors may visualize non-sporulating cells that obviously still active and motile. The anticorrelation between motile and sporulating cells could be easily demonstrated using double reporter strains.

In general, I do not agree with the strong statement that escape mechanisms are activated, unless the authors unequivocally demonstrate that any of these mechanism support escape and survival of the Dmatrix strains in the presence of PCL. Until than, the chapter title is misleading.

Line 386: is the delayed sporulation due to specific induction of sporulation by competition or simply due to nutrient availability due to lack of PCL growth! These two should be experimentally verified, or the claim should be removed. If I understand correctly, colonization pattern by 3610 and Dmatrix is comparable and does not depend on PCL?

The transcriptome experiment on *B.subtilis* (at least those cells that did not sporulated) shows that about 1/4th of the genome is differentially expressed. How can the authors exclude the possibility that other processes are responsible for the increased sensitivity of Dmatrix genes and enhanced sporulation compared to 3610? No actual experiment is presented that shows that biophysical properties are important for protection.

Other comments:

I do not understand the last sentence of the abstract

Line 62 overly complex sentence: "a combination of ... including ... in conjunction with the ..."

Line 101: previous studies showed that genes are expressed differentially, but no study has addressed that indeed matrix components are differentially produced during these interactions. Therefore, sentence should be rewritten to highlight transcriptional change, but not the matrix components itself

Line 109: please cite publication that *B.subtilis* and *P.chlororaphis* coexists in any sample otherwise remove this statement, or modify that both has been shown to colonize plants (still direct citation is needed that demonstrated the presence of these species in the rhizosphere microbiome, not simply as plant/soil additive)

Line 118: include growth curves in LB medium for both strains to support claim on comparable growth rates (as supplementary figure)

Line 131: correct *bsla* to *bsIA*

Fig 1b: Seminara and colleagues have reported that *eps* mutant colony is smaller than 3610 wild type, while the authors present a possibly bigger colony for the matrix mutant (probably, lack of scale bar circumvent clear conclusions). Also, results in line 201 similarly suggest faster spreading of matrix mutant.

Line 162: the protective nature of *Bacillus* spores in case of *Myxococcus* has been demonstrated before and can be cited here as a motivation why to assay spore numbers

Line 211: the initiation of sporulation could be very easily tested by repetition of the experiments with a sporulation specific reporter

Line 231: data should be presented as supplementary information (data not shown nowadays is unusual, especially in the case of *Nat Commun*)

Line 265: which experiment demonstrates the structural weakness? Rheology should be performed to demonstrate structural differences among strains under the applied conditions.

Line 371: provide reference for both bacteria

Line 376: what is the colonization rate of *Dmatrix* compared to 3610 when inoculated alone?

Line 380: what is "co-bacterization"?

Line 394: the term "related" suggest genetical relatedness, please replace

Line 403 and 483: why is this interaction social? Please explain

Line 470: the reason might be simply nutritional differences due to altering matrix defense or T6SS competition

Line 474: the impact of spatial distribution/assortment of strains is not shown in vitro or in vivo. It might be an indirect outcome of competition. The sentence currently suggests the impact of de novo established spatial distribution on competition

Line 479: the two factors were not dissected in the manuscript. Competition might induce differentiation, as has been demonstrated by other studies.

Line 483: why are these adaptive strategies? No experiment is presented that adaptation (i.e. genetic changes via evolution) is affected during interaction.

Line 487: no evidence is shown in the manuscript that these traits evolved for multispecies community living. Also, please explain how this knowledge can be used to change agricultural practices? These last sentences are outreaching to evolutionary theory and agricultural applications, but no direct guidelines or suggestions are provided, therefore these could be omitted.

Line 493: explain LB, i.e. lysogeny broth

Reviewer #2 (Remarks to the Author):

This manuscript describes an interaction between *Bacillus subtilis* and *Pseudomonas chloroaphis* (PCL 1606), two organisms found in the soil that are relevant to plant health. They showed that the extracellular matrix of *B. subtilis* protects cells from killing by *P. chloroaphis* when the two species were grown in proximity. They then used time-lapse microscopy to show the interaction on a cellular scale, which highlighted killing of *B. subtilis* matrix mutant. Positives of the paper include the use of this novel time-lapse microscopy technique to image colony interactions, the effort to dissect an environmentally relevant bacterial interaction, and the direct testing of the interaction on melon leaves. Overall, however, the data in this paper are disconnected and many findings are not properly followed up on or do not include critical controls (see below). In addition, the figures were extremely hard to interpret and many had poor or missing labels; it was unclear in many instances to determine what I was looking at or which data corresponded to which strains.

Major comments:

1. Title: The title is misleading, and an inaccurate representation of the data shown in the paper. There is little data to support the role of T6SS in this coculture interaction. To begin, the authors never investigated the relevance of the T6SS system in the context of the wildtype *B. subtilis* interaction, nor were its transcription levels examined in the wildtype coculture (i.e. using RT-PCR). However, these experiments are potentially irrelevant: from what I can see, it appears that the T6SS mutant has essentially the same phenotype as the wildtype *P. chloroaphis*: there is still invasion and the CFUs look similar/almost identical (Fig. 2 (WT) and Fig S8 (T6SS mutant)). It is unclear to me why the authors propose that T6SS has an important role in this phenotype based on these data.

2. There were multiple instances when genes were proposed to be relevant to this interaction with weak evidence to support them:

a. In terms of surfactin, the IMS data (Fig. 4a-d) are not compelling.

1) There may not actually be a 'concentration increase' in the center of the matrix mutant colony compared to wildtype, just less surfactin being made overall – showing a heatmap of peak intensity (rather than a single-color distribution map) would clarify this. (It is possible/likely that in the monoculture matrix mutant the center of the colony simply has more surfactin than the agar (which would be clear in a heatmap image), and so when the overall number of viable cells are reduced in the coculture, the surfactin appears primarily in the colony rather than in the agar.)

2) Why was a surfactin mutant not examined? The production and localization of surfactin in both

wildtype and matrix mutant of *B. subtilis* does not allow for any conclusions to be drawn about the movement of *P. chloroaphis* without surfactin mutant coculture data.

3). In addition, what is the expansion rate of PCL 1606 in an interaction with a *B. subtilis* *srfAA* mutant compared to its interactions with *B. subtilis* 3610 wild-type? I'm not convinced that surfactin is mediating the spread of the colony. Does PCL 1606 expand in the presence of *B. subtilis* matrix mutant conditioned media?

Without these data it is difficult to understand a claim that surfactin is relevant to this interaction.

b. Why was the relevance of the T6SS not examined in the wildtype *B. subtilis* interaction (Fig. 5)? Is T6SS gene expression occurring before cell-cell contact? See above note (in title section) about the *P. chloroaphis* T6SS mutant. If T6SS was important to the interaction, we would expect less *B. subtilis* killing and an inability to infiltrate the colony. I would suggest comparing the intermediate region in a *B. subtilis* wild-type- *P. chloroaphis* and *B. subtilis* matrix mutant by RNA-seq to determine the genes that encode products important to this interaction.

c. The microscopy visualizing the *B. subtilis* motility reporter does not seem relevant to the story (Fig. 6). It is again in a *B. subtilis* matrix mutant background and no biological controls are shown. A more relevant question to pursue would be why the extracellular matrix mediates resistance to *P. chloroaphis* invasion, which could follow after Figure 3.

3. There were multiple issues with the Figures and how the data was presented.

- Figure 1: It's difficult to observe the invasion in these images. Can you add an inset with a close-up of the intermediate? Also, please label the colonies directly rather than just with "a,b,c,d,e" – it makes unnecessary work for your reader to have to constantly look back at the legend to figure out what strains are being shown and to interpret the data – simply put the labels on the image.

- Figure 2: What are a, b, c? Are they I, II, III? Not explained anywhere. Also, a,b,c appear to be identical data to d,e,f – why are both being shown?

- Figure 4:

- a. See above. It seems to me that the *B. subtilis* matrix mutant would be producing less surfactin during the interaction because less cells *B. subtilis* cells are present. Or is surfactin truly downregulated by *B. subtilis* in the interaction?

- b. Does PCL 1606 produce a molecule with surfactant properties that may have a different m/z? There are many lipopeptides generated by pseudomonads that might have similar chemical characteristics in terms of facilitating cell spreading

- c. Panel f – what the two colors correspond to in terms of strains did not appear to be mentioned anywhere.

- Figure 5: Panels b – g are uninterpretable. I have no idea what I am looking at in these images. Some context or orientation for the reader is necessary.

- Figure 6: The microscopy with the motility reporter does not seem relevant to the story, as noted above.

- Figure 7cd: Indicate leaf surface. (Again, provide some context in terms of a DIC/brightfield image/diagram/etc. to explain what this image is showing.) Does the white color indicate overlap between strains? Again, this is not mentioned.

4. I have serious reservations about all of the CFU data. From what I can tell from the methods, the cells were merely "completely resuspended" (line 558) rather than being sonicated prior to plating for CFU counts. It is well established that *B. subtilis* frequently forms cell chains, and sonication is required to separate these chains into individual cells. In the absence of this experimental step, all of the CFU counts could be a dramatic underestimation (i.e. a single chain with many dozens or hundreds of cells would lead to a single apparent CFU; thus the CFU quantification may not accurately represent CFU from single cells.)

Minor comments:

- Keep consistent with strain names (i.e. choose one to use after introducing the strain). There are multiple places where *Pseudomonas* and *P. chloraphis* are used interchangeably with PCL 1606.
- Line 48: Change “metabolic consortia” to “metabolic exchange”
- Line 52: Change “they” to “it”
- Line 58-59: Change “little is known about their functional importance in real environments” to “little is known about the functional importance of extracellular matrix components in natural environments”
- Line 77: Change “cohabit” to “coexist”
- Lines 82-96: Changing between past and present tense
- Line 94: Change co-bacterizations to co-colonization of bacteria
- Line 96: Remove determinant
- Line 96: Change “plant seeds germination” to germination of plant seeds
- Line 100: Write out *Pseudomonas*
- Lines 101-109 belong in the Introduction
- Figure 1: It’s difficult to observe the invasion in these images. Can you add an inset with a close-up of the intermediate?
- Supp Figure 1d. No wrinkles present on *B. subtilis* 3610 after 72h growth on MSgg. What temperature are these grown at?
- Line 131: Change “bsla” to “BslA”
- Line 154: Remove “in the interaction between the two wild-type strains” because it led me to think you were talking about the intermediate area
- Lines 164-165: List percentages of sporulation in the same order for an easier comparison for your reader
- Line 168: Remove “secondary”
- Line 199: Remove “very”
- Line 226: Replace “To clarify this moment of the interaction” with “To gain insight into factors that mediate the observed colony expansion rates”
- Figure 4: Does PCL 1606 produce a molecule with surfactant properties that may have a different m/z?
- Line 319: Change “spo0A” to “Spo0A”
- Line 320: Change “interaction” to “intermediate” for consistency
- Line 346: Correct “give rises” to “give rise”
- Figure 7cd: Indicate leaf surface. Does the white color indicate overlap between strains?
- Line 347: Define “radicle” for those unfamiliar with this plant biology term
- Lines 411-414: Rewrite sentence in parallel “More specifically, BslA and exopolysaccharide contribute hydrophobicity and resistance to diffusible active molecules, respectively, to *B. subtilis* biofilms
- Line 458: Remove “would be inflicted by” and replace with “is”
- Line 461: Replace “proven” with “shown”
- Line 453: Replace “to be depended on” with “seems to depend on”
- Lines 455-456: Remove “and the maintaining of a small sub-population able to move and escape from the *Pseudomonas* presence by the activation of motility elements.” If removing data
- Line 646: When was LC-MS/MS used in the study?
- Line 1006: State what color line represents *B. subtilis* and PCL 1606
- Line 1009: Unclear what the term spatially distributed means
- Line 1021: Add that *B. subtilis* is labeled using CFP to figure legend

Reviewer #3 (Remarks to the Author):

[The editors asked this reviewer to comment in particular on the MALDI-IMS imaging analyses.]

Overall, the manuscript is clearly written. The work is easy to understand, even for others in a different field. In this report, the authors explored the function significance of biofilms during the interactions between 3610 and PCL 1606. This study demonstrated how the extracellular matrix protects *B. subtilis* colonies from the infiltration by *Pseudomonas*, using microscopy, RNA-seq analysis, and mass spectrometry imaging.

Specific comments:

In supplemental figure 1, the existence of a "clear" halo is not evident, specifically in (c). Can you add arrows or labels indicating the halo for this image?

In line 142-146: It is difficult to find the labels I and II indicated in Fig. 2, 3, and Suppl. Fig 3. Can you add more description to your text (i.e. Fig. 2g-I) so it can be easy to find?

Consistency: In the caption of Suppl. Fig 3., the text compares an empty bar with a black bar, while in caption Suppl. Fig 4, the text compares a white bar vs. a black bar. Make sure to keep the text and explanation consistent. The same occurs with Fig. 5.

For Fig 3: What is the scale of your white scale bar?

MSI related comments

Is there a reason why you are not including metabolites produced by *B. subtilis* that show similar distribution patterns in both interactions and monocultures in your Suppl. Fig. 5? Show some MS images of metabolites that actually have signal.

Is there a reason why you are only showing $[M+Na]^+$ MS images, while the molecular network results are of $[M+H]^+$? How does the data compare between the different adducts? Do you observe other adducts in the MS data (i.e. K^+)? Have you tried imaging the summed adduct intensity and doing the same for the molecular network?

How did you generate and compare the MS images? For example, are you generating the image based on signal intensity, ratio to internal standard, TIC, etc?

In line 278-280, the text indicates that the main transcriptional changes can be categorized in (i) amino acid and carbon metabolism. In the MS data, did you see any metabolites that correspond to the amino acid and carbon metabolism?

How did you confirm the identity of C14-surfactin and C15-surfactin peaks in MS?

In line 637, you state that CHCA/DHB is a universal matrix. Be careful when claiming a matrix as "universal" since that does not exist.

Reviewer #1 (Remarks to the Author):

The manuscript present an interesting observation: colonies of *Bacillus subtilis* and *Pseudomonas chlororaphis* show a defined interaction pattern which is altered if *B.subtilis* is lacking matrix production. In a series of in vitro LB plate based experiment, the authors demonstrate that deletion of the 3 matrix component has great impact on sporulation in the colonies, spatial distribution of surfactin, while inactivation of T6SS in *Pseudomonas* alters this interaction. Finally, they show that matrix deletion in *Bacillus* or T6SS deletion in *Pseudomonas* also alters the interaction and mostly the sporulation efficiency when inoculated on melon leaves or seeds. There are lots of very interesting information in the manuscript which all suggest how the two species interact.

There are few grammar mistakes in the manuscript. In general, the manuscript might require a professional text/grammar editor so the storyline can be easier followed.

R - We greatly appreciate the time and effort the Reviewer has spent reviewing the manuscript, their careful criticism and constructive comments. We have tried to improve the readability of the manuscript as a whole and the finalized text has been carefully revised and corrected by one of the authors, Dr. John R. Pearson, a native English speaker. Below we list our responses to the Reviewer's specific comments.

Major comments:

Q - All colony images are lacking scale bars and should be included in each figures with colony images.

There are some inconsistencies in the colony sizes among images. For example, fig 1d PCL colony is seems smaller at 96h compared to 72h, similarly fig 4c and d: PCL at 48 h seems smaller than 24 h samples. How consistent were the colony sizes? Is this due to the use of a normal camera instead of a zoom microscope, where magnification (and therefore also scalebar) can be controled and recorded.

R - In order to improve consistency between figures and to make the progression of the bacterial interactions easier to follow, we have taken new pictures using a stereo microscope with scale bars included for all images. Colony sizes in Fig. 1 and Suppl. Fig. 3 are now consistent with the colony ages. Furthermore, and in accordance with related comments from Reviewer 2, we have modified Fig. 1 to include higher magnification images of the intermediate area of the interactions. Control colonies have been relocated to Suppl. Fig. 3.

Q - Fig 2a-e seems to fully represent the figure panels in Supplemental Fig 8a-e - why is this the case? This suggest hardly any impact of T6SS mutation on the interaction with the Dmatrix strain

R - We wish to thank Reviewer 1 very much for spotting this issue. As also noticed by reviewer 2, Suppl. Figure 8 (now Fig. 7) was identical to Fig. 2. This was due to an extremely unfortunate mistake made during data processing and preparation of the original figure. We have corrected these errors and made sure the graphs correctly reflect the data we originally obtained. The corrected graphs are now included as Fig. 7.

As indicated in the original manuscript, the invasion patterns, in general, are similar in the interaction of Δ matrix with PCL1606 or with Δ T6SS. However, in terms of sporulation rates, our data shows that the Δ T6SS mutation significantly decreases the sporulation rate of Δ matrix compared to interaction with the wild type PCL1606 (Fig 2b and Fig. 7b). These results concur with those obtained in plants, where we see a role for T6SS in the increased sporulation of *Bacillus* strains (See Figures 8 and 9).

To give a more complete picture of this interaction we have included new *in vitro* experiments, which demonstrate the existence of second sporulation induction mechanism that occurs prior to contact between the species, which depends on the kinase KinB (Suppl. Fig. 7), suggesting the involvement of environmental and nutrient cues (See also the response below related to the use of the *PsspB-yfp* reporter gene).

Thus our suggestion is that at least two different pathways are involved in inducing *Bacillus* sporulation; a KinB dependent long-range mechanism and a T6SS-dependent mechanism that depends on the direct cell-cell contact that occurs during infiltration of Δ matrix.

The new results have been included at lines 161-169 and are discussed at lines 508-515.

Q - Is the T6SS activated when in contact with 3610 strain? Reporter measurement is missing for the combination of PCL and 3610. Also, does the interaction change between 3610 and *tssA* mutant PCL?

R - To address these points we have added a number of new experiments to the manuscript. Firstly, we have documented the interaction between the T6SS mutant and wildtype 3610 with images and CFU/ml counts (Suppl. Fig. 19), which shows no difference with respect to the wildtype PCL1606 strain. Secondly, using the methodology developed for the timelapse assays, we have measured T6SS promoter expression using PCL1606 carrying a P_{T6SS} DsRed construct at 72h during interactions between both wildtype and Δ matrix. We find that T6SS is much more weakly induced during interaction with wildtype 3610 than in the case of the Δ matrix (new Fig. 6). These new results are described in the Result section (Lines 343-349) where we argue that these data strongly support a role for the extracellular matrix in preventing close cell-cell contact between the two species.

Q - Generally, it is well established that matrix non-expressing cells show high motility (e.g. *ymdB* mutant lacking matrix production). Thus it is hardly surprising that motile cell clusters are detectable. I am missing a proper control to demonstrate that in the absence of PCL, motility is not expressed highly in the Δ matrix strain. The authors may visualize non-

sporulating cells that obviously still active and motile. The anticorrelation between motile and sporulating cells could be easily demonstrated using double reporter strains.

In general, I do not agree with the strong statement that escape mechanisms are activated, unless the authors unequivocally demonstrate that any of these mechanism support escape and survival of the Dmatrix strains in the presence of PCL. Until then, the chapter title is misleading.

R - We thank the reviewers for their highly valid comments related with *B. subtilis* motility. To address these points, we have performed new experiments in an attempt to shed more light on whether the expression of motility-related genes could constitute an escape strategy. We find that increased motility gene expression is related to contact with PCL1606 and not merely due to loss of the extracellular matrix. In addition, new RNAseq data from the interaction PCL1606-3610 (see also response to reviewer 2) also show motility and flagellar induction in the intermediate area. Furthermore, our RNAseq data comparing 3610 and Δ matrix single colonies show an induction of motility in Δ matrix [Redacted], supporting our hypothesis. However, we have not been able to obtain sufficient evidence to directly prove the existence of escape mechanisms induced by contact with PCL1606. Given that this experiment is not part of the paper's core focus on the extracellular matrix, we propose removing explicit references to the existence of a *Bacillus* escape mechanism, while retaining references to changes in motility-related gene expression with modifications to Results section (lines 368-389).

[Redacted]

Q - Line 386: is the delayed sporulation due to specific induction of sporulation by competition or simply due to nutrient availability due to lack of PCL growth! These two should be experimentally verified, or the claim should be removed. If I understand correctly, colonization pattern by 3610 and Dmatrix is comparable and does not depend on PCL?

R - In agreement with the reviewer's comment, delayed sporulation could be related to many factors including nutrient availability. Indeed, our new finding that the "contactless" induction of sporulation in 3610 is dependent on KinB suggests that in some cases changes in nutrient availability may play an important role. However, our finding in melon leaves and seeds that the loss of T6SS results in a reduction from 70% to 30% sporulation at 2 dpi despite similar overall CFU counts (suggesting similar levels of nutrient availability) argues that direct competition plays a dominant role where there is direct contact between the bacteria.

CLSM images of the colonization pattern of single *Bacillus* inoculations in melon leaves have been included in the manuscript (see Suppl. Fig. 23). As correctly understood by the reviewer, the colonization pattern of both *Bacillus* strains assayed are comparable, and are similarly distributed on leaves in the absence of PCL1606.

Q - The transcriptome experiment on *B. subtilis* (at least those cells that did not sporulated) shows that about 1/4th of the genome is differentially expressed. How can the authors exclude the possibility that other processes are responsible for the increased sensitivity of Dmatrix genes and enhanced sporulation compared to 3610? No actual experiment is presented that shows that biophysical properties are important for protection.

R - As correctly mentioned by the reviewer, transcriptomic experiments showed important transcriptomic changes and indeed we cannot exclude the involvement of other processes in the interaction. Evidently, experiments comparing whole transcriptomes are likely to identify a very large number of genes with altered expression. However, it is impractical to test each and every differentially expressed gene. Therefore, in this paper we have focused in a few pathways that, according to our data, may contribute to the initial macroscopic phenotypes of the interaction.

It is extremely difficult to definitively demonstrate that altered biophysical properties are solely responsible for a given change but we have attempted to better characterize the biophysical properties in the presence and absence of the matrix genes to see if they correlate with their altered resistance to invasion. By performing high resolution Magnetic Resonance Imaging and solid state NMR we measured a higher water diffusion rate in Δ matrix compared to 3610, a parameter inversely proportional to bacterial colony compaction, especially in the front line of colony. We also found that the Δ matrix colony is more homogeneous than 3610 in terms of water availability, making the Δ matrix colony fluid. In addition, solid state NMR data support MRI data at the atomic level and highlight differences in the composition of the colonies. While these results do not exclude the involvement of other factors, they strongly suggest a strong correlation between these altered properties and invasion susceptibility.

The new data has been included in the Result section (lines 217-261), Discussion (lines 468-473) and Fig. 4.

Q - Other comments:

I do not understand the last sentence of the abstract

R - We have re-written the sentence to clarify the message (lines 39-40).

Q - Line 62 overly complex sentence: "a combination of ... including ... in conjunction with the ..."

R - This has been corrected.

Q - Line 101: previous studies showed that genes are expressed differentially, but no study has addressed that indeed matrix components are differentially produced during these interactions. Therefore, sentence should be rewritten to highlight transcriptional change, but not the matrix components itself.

R - We have rewritten the sentence, thank you for the suggestion. Now it states: "Previous studies have reported differential transcriptional control of matrix component expression in interactions between *Bacillus* and other bacterial species".

Q - Line 109: please cite publication that *B.subtilis* and *P.chlororaphis* coexists in any sample otherwise remove this statement, or modify that both has been shown to colonize plants (still direct citation is needed that demonstrated the presence of these species in the rhizosphere microbiome, not simply as plant/soil additive)

R - We have modified the sentence to avoid misunderstanding. Although we don't have reports indicating co-existence of these species in soil samples, their places of isolation and the properties of the two bacterial species used in this study have led us to hypothesize their potentially coexistence in the same environments. We have clarified the sentence to indicate that both can be found in plant-related environments (PCL1606 was isolated from avocado rhizosphere and many *B. subtilis* strains are frequently found in plant microbiomes and soil). We have cited in the manuscript the paper in which PCL1606 was firstly described.

Q - Line 118: include growth curves in LB medium for both strains to support claim on comparable growth rates (as supplementary figure)

R - Growth curves in LB medium have been performed and added as Suppl. Fig. 2. As can be observed, the growth dynamics of *Bacillus* and *Pseudomonas* in LB medium are similar.

Q - Line 131: correct *bsla* to *bslA*

R - This change has been incorporated.

Q - Fig 1b: Seminara and colleagues have reported that *eps* mutant colony is smaller than 3610 wild type, while the authors present a possibly bigger colony for the matrix mutant

(probably, lack of scale bar circumvent clear conclusions). Also, results in line 201 similarly suggest faster spreading of matrix mutant.

R - The reviewer is correct in their observations. However, it is important to note that Seminara and colleagues performed their experiments using Mmsg medium while we used LB. In LB, the wild type 3610 colony is smaller than the Δ matrix mutant colony. In fact, as indicated in line 199, Δ matrix spreads faster than wild type, an effect that we believe is directly related with the increased fluidity and water availability of Δ matrix as demonstrated by MRI and solid state NMR. We have included scale bars in the pictures to make these type of comparisons easier.

Q - Line 162: the protective nature of Bacillus spores in case of Myxococcus has been demonstrated before and can be cited here as a motivation why to assay spore numbers

R - That is totally right. The appropriate reference has been included in the text (See line 155).

Q - Line 211: the initiation of sporulation could be very easily tested by repetition of the experiments with a sporulation specific reporter.

R - Following the reviewer's suggestion, we have performed pairwise time-lapse interaction assays using a Δ matrix strain carrying a construct expressing YFP under the control of the *sspB* promoter, a gene previously used as a sporulation specific reporter gene.

These new results confirm that sporulation is induced before the first direct contact with PCL1606, what is in accordance with the spore count assays shown in Fig. 2. We also observed a gradual decrease in P_{sspB} -YFP fluorescence after induction, mostly likely indicating entry into sporulation. These observations have been included in the manuscript (See lines 206-211) and in Suppl. Fig. 8.

Q - Line 231: data should be presented as supplementary information (data not shown nowadays is unusual, especially in the case of Nat Commun).

R - Following the reviewer's suggestion, we have added supplementary figures (Suppl. Fig. 9, 10 and 11) showing the MALDI-TOF MSI patterns of each experiment performed at 24 h, 48 h and 72 h, respectively. As can be observed, PCL1606 metabolite patterns are similar in single colony and pairwise interaction assays. In addition, we have added MALDI-TOF MSI images of some representative PCL1606 compounds (Suppl. Fig. 12) and *Bacillus* strains (Suppl. Fig. 13) when growing alone or in interaction.

Q - Line 265: which experiment demonstrates the structural weakness? Rheology should be performed to demonstrate structural differences among strains under the applied conditions.

R - As correctly pointed by the reviewer, no experimental results were shown in the original submitted version of the manuscript, although visual changes were evident comparing 3610 to Δ matrix colonies. As described above we have done experiments to decipher their biophysical properties and how these changes are linked to protecting 3610 colonies against PCL1606 invasion (See new section in lines 217-261 and Fig. 4). Our new results obtained by MRI and NMR confirm that the absence of extracellular matrix modifies the composition and structure of colonies in terms of bacterial density/fluidity, water content and colony organization and composition. These observations are strong evidence of structural weaknesses in Δ matrix colonies being the major cause of their inability to defend against PCL1606 invasion.

Q - Line 371: provide reference for both bacteria

R - Done, thank you.

Q - Line 376: what is the colonization rate of Δ matrix compared to 3610 when inoculated alone?

R - The colonization rate of Δ matrix is slightly lower than 3610, although this difference is not statistically significant. This result is shown in Fig. 9a.

Q - Line 380: what is "co-bacterization"?

R - The term co-bacterization refers to experiments where leaves or seeds were inoculated with PCL1606 and 3610 strains simultaneously. To do so, we grew each bacterium separately overnight, washed them twice and resuspended them in water. After that, bacteria were mixed and applied together to leaves and seeds, as indicated in the Materials and Methods (See lines 771-778).

Q - Line 394: the term "related" suggest genetical relatedness, please replace

R - This has been corrected.

Q - Line 403 and 483: why is this interaction social? Please explain

R - The term 'social interaction' refers to the different behaviors adopted by bacteria in response to other microorganisms found in the same environment. In the case of this study, we observe the bacterial responses to the presence of other microorganisms. In our opinion, this interaction can be understood as social.

Q - Line 470: the reason might be simply nutritional differences due to altering matrix defense or T6SS competition.

R - Yes, we believe both mechanisms are involved depending on the circumstances. As described above, re-analysis of the corrected data shown in the new Fig. 7, explained in the manuscript (Lines 357-362), together with new data presented in Suppl. Fig 7 and lines 161-169, suggest an initial KinB-dependent sporulation induction, a factor involved in sensing nutritional and environmental changes, followed by contact-dependent induction that requires a functional PCL1606 T6SS.

Q - Line 474: the impact of spatial distribution/assortment of strains is not shown *in vitro* or *in vivo*. It might be an indirect outcome of competition. The sentence currently suggests the impact of *de novo* established spatial distribution on competition

R - Thank you for this comment. We agree with the reviewer. We have deleted the sentence related with spatial distribution to avoid misunderstanding.

Q - Line 479: the two factors were not dissected in the manuscript. Competition might induce differentiation, as has been demonstrated by other studies.

R - We have modified this sentence to clarify the main role of the T6SS in triggering sporulation *in vivo* and *in vitro* as stated above (See lines 511-515).

Q - Line 483: why are these adaptive strategies? No experiment is presented that adaptation (i.e. genetic changes via evolution) is affected during interaction.

R - The reviewer is right. We have deleted the term "adaptation" from the manuscript in order to avoid confusion.

Q - Line 487: no evidence is shown in the manuscript that these traits evolved for multispecies community living. Also, please explain how this knowledge can be used to change agricultural practices? These last sentences are outreaching to evolutionary theory and agricultural applications, but no direct guidelines or suggestions are provided, therefore these could be omitted.

R - We do believe our findings are relevant for the areas mentioned. In the paper, we describe an *in vitro* “negative” interaction between bacterial species, where we might conclude that they are acting antagonistically. On the other hand, we show a very different type of interaction on leaves where they appear to co-exist in a more stable situation. In both cases the presence of an extracellular matrix affects their interaction but in different ways. The relevance of this particular bacterial interaction in real environments remains to be tested but illustrates the complexity involved in bacteria-bacteria and bacteria-plant interactions.

What we were trying to highlight in the discussion was that negative interactions in one situation do not necessarily mean that the same species cannot cooperate in another one, and that genetic mechanisms may have evolved to promote such cooperation. We also think it is worthwhile suggesting that manipulating the extracellular matrix or pre-inoculating seed with different bacterial combinations might be effective methods for changing the organization of plant-associated bacteria in order to enhance their beneficial traits.

We have modified the Discussion (Lines 531-537) to better explain these points and have included practical examples of how the ideas mentioned could have real world applications. However, if the Reviewer and Editor think appropriate we would be prepared to remove these topics from the Discussion.

Q - Line 493: explain LB, i.e. lysogeny broth

R - Done. LB refers to Luria-Bertani broth.

Reviewer #2 (Remarks to the Author):

This manuscript describes an interaction between *Bacillus subtilis* and *Pseudomonas chloroaphis* (PCL 1606), two organisms found in the soil that are relevant to plant health. They showed that the extracellular matrix of *B. subtilis* protects cells from killing by *P. chloroaphis* when the two species were grown in proximity. They then used time-lapse microscopy to show the interaction on a cellular scale, which highlighted killing of *B. subtilis* matrix mutant. Positives of the paper include the use of this novel time-lapse microscopy technique to image colony interactions, the effort to dissect an environmentally relevant bacterial interaction, and the direct testing of the interaction on melon leaves. Overall, however, the data in this paper are disconnected and many findings are not properly followed up on or do not include critical controls (see below). In addition, the

figures were extremely hard to interpret and many had poor or missing labels; it was unclear in many instances to determine what I was looking at or which data corresponded to which strains.

R - We would like to express our gratitude for the reviewer's critical evaluation of our manuscript. We have taken on board their suggestions and believe that the revised submission has been greatly improved as a result. We have done our best to improve the readability of the manuscript, make the figures easier to understand and generally improve the connectedness and flow of the manuscript. Below we list our responses to the specific comments of the referee.

Major comments:

Q - 1. Title: The title is misleading, and an inaccurate representation of the data shown in the paper. There is little data to support the role of T6SS in this coculture interaction. To begin, the authors never investigated the relevance of the T6SS system in the context of the wildtype *B. subtilis* interaction, nor were its transcription levels examined in the wildtype coculture (i.e. using RT-PCR). However, these experiments are potentially irrelevant: from what I can see, it appears that the T6SS mutant has essentially the same phenotype as the wildtype *P. chloroaphis*: there is still invasion and the CFUs look similar/almost identical (Fig. 2 (WT) and Fig S8 (T6SS mutant)). It is unclear to me why the authors propose that T6SS has an important role in this phenotype based on these data.

R - We agree that the title did not correctly reflect the results presented in the original manuscript and may have been misleading, although that was not our intention. To address these issues we have made a series of changes to the article.

Firstly, we must sincerely apologize for an extremely unfortunate error in the preparation of one of the graphs in the original paper (the old Suppl. Fig. 8) where we inadvertently used the wrong dataset, resulting in the graph being the same as the one presented in Figure 2, a problem also spotted by Reviewer 1. We have corrected this mistake and the revised graph is now included as Fig. 7. Now, when comparing the overall invasion dynamics of Δ matrix by wildtype PCL1606 (Suppl. Fig. 19) or Δ T6SS (Fig. 7), they look quite similar. However, in terms of sporulation rates, when comparing Fig 2b and Fig. 7b, sporulation sharply decreases in the interaction of Δ matrix with Δ T6SS at 96h. This data shows that T6SS does have a role inducing sporulation during later stages of the interaction with Δ matrix.

Next, we have included new experiments, shown in Suppl. Fig. 7, that demonstrate that the histidine kinase B is the main receptor involved in sensing signals that trigger sporulation before the two colonies come into contact. This kinase has been proposed to be sense low nutrient levels and environmental changes.

In addition, we have studied the relevance of the T6SS in a 3610 context. To address this point, we have added several new experiments to the manuscript. Firstly, we have documented the interaction between the T6SS mutant and wildtype 3610 with images and CFU/ml counts (Suppl. Fig. 19), which shows no difference with respect to the wildtype

PCL1606 strain. Secondly, as suggested by the reviewer (see below in Page 3 of reviewer's responses), new RNAseq data obtained from the PCL1606 – 3610 interaction show differential expression of T6SS-related genes at 72 h of interaction, when PCL1606 and 3610 are in contact between their respective colony leading edges. Thirdly, using the methodology developed for the timelapse assays, we have measured T6SS promoter expression using PCL1606 carrying a P_{T6SS} DsRed construct at 72h during interactions between both wildtype and Δ matrix 3610. We find that T6SS is much more weakly induced during interaction with wildtype 3610 than in the case of the Δ matrix (new Fig. 7). These new results are described in the Result section (Lines 343-349) where we argue that these data strongly support a role for the extracellular matrix in preventing close cell-cell contact between the two species.

Thus, we propose that at least two different pathways are involved in inducing *Bacillus* sporulation; a KinB dependent long-range mechanism activated before close contact with PCL1606, and a T6SS- dependent mechanism that depends on the direct cell-cell contact that occurs during infiltration of 3610 Δ matrix.

We have modified the text to reflect these new results at lines 161-169, 350-362 and 508-515 together with new or modified Fig. 7, 8, Suppl. Fig. 7 and 19. Moreover, in response to the referee's comments and our new data we have also modified the title to remove the overly strong claim for T6SS dependency.

Q - 2. There were multiple instances when genes were proposed to be relevant to this interaction with weak evidence to support them:

a. In terms of surfactin, the IMS data (Fig. 4a-d) are not compelling.

1) There may not actually be a 'concentration increase' in the center of the matrix mutant colony compared to wildtype, just less surfactin being made overall – showing a heatmap of peak intensity (rather than a single-color distribution map) would clarify this. (It is possible/likely that in the monoculture matrix mutant the center of the colony simply has more surfactin than the agar (which would be clear in a heatmap image), and so when the overall number of viable cells are reduced in the coculture, the surfactin appears primarily in the colony rather than in the agar.)

R - We have prepared new MALDI-TOF MSI figures using a heatmap scale instead of single-color distribution maps (See Fig. 6). The new images confirm our original observations that higher concentrations of surfactin are present in the center of the Δ matrix colony than in 3610. Furthermore, we have added LC-MS/MS traces with the different surfactin isoforms produced in the interaction PCL1606 vs 3610 or PCL1606 vs Δ matrix (Suppl. Fig. 15).

Q - 2) Why was a surfactin mutant not examined? The production and localization of surfactin in both wildtype and matrix mutant of *B. subtilis* does not allow for any

conclusions to be drawn about the movement of *P. chloraphis* without surfactin mutant coculture data.

R - We fully agree that it would be highly informative to examine surfactin mutants in this context. Therefore, we have generated a quadruple mutant (Δ matrix Δ surf) carrying an interrupted *SrfAA* gene in a Δ matrix background, with the idea to get more insights into the role of surfactin in PCL1606 spreading and in the invasion of Δ matrix colonies (remember that invasion does not happen against 3610 WT). The results obtained are quite interesting and support the data previously shown in the manuscript. Estimations of the bacterial population (cfu/ml) in the Δ matrix Δ surf-PCL1606 interaction showed a clear delay in the penetration of PCL1606 into the Δ matrix Δ surf colony (See Fig. 5e). In this new figure we have compared the percentage of Δ matrix population and Δ matrix Δ surf population in the *Bacillus* area (BA) after 72h of interaction, and results show that the percentage of *Bacillus* cells is significantly higher in the case of Δ matrix Δ surf, indicating a reduction of the PCL1606 population in this area of the interaction in the absence of surfactin. See also our answer to the next comment related to the spread of PCL1606. These results have been included and discussed in the manuscript (Results section lines 298-306, Discussion 486-492 and Fig. 5e).

Q - 3). In addition, what is the expansion rate of PCL 1606 in an interaction with a *B. subtilis* *srfAA* mutant compared to its interactions with *B. subtilis* 3610 wild-type? I'm not convinced that surfactin is mediating the spread of the colony. Does PCL 1606 expand in the presence of *B. subtilis* matrix mutant conditioned media? Without these data it is difficult to understand a claim that surfactin is relevant to this interaction.

R – For the experiment presented in the previous version of the manuscript we used pure surfactin rather than *B. subtilis* matrix mutant conditioned media to avoid the difficulty in interpreting the effects of a much more chemically complex situation.

For the new manuscript, we have performed additional timelapse microscopy experiments comparing interactions between PCL1606– Δ matrix Δ surf, PCL1606– Δ matrix and PCL1606 growing alone. In accordance with the data observed previously (Fig. 2f and Fig. 5f), the main changes to PCL1606 colony expansion occur during the first ~10 hours of the interaction. Mean expansion rates for each interaction or PCL1606 growing alone suggest that surfactin can influence PCL1606 cell spreading. Thus, chemical, biological and microscopy data all suggest a role for surfactin in facilitating the spreading of PCL1606, although this does not exclude the participation of additional factors. These results have been added and discussed in the manuscript (See Lines 298-306).

Q - b. Why was the relevance of the T6SS not examined in the wildtype *B. subtilis* interaction (Fig. 5)? Is T6SS gene expression occurring before cell-cell contact? See above note (in title section) about the *P. chloraphis* T6SS mutant. If T6SS was important

to the interaction, we would expect less *B. subtilis* killing and an inability to infiltrate the colony. I would suggest comparing the intermediate region in a *B. subtilis* wild-type- *P. chloroaphis* and *B. subtilis* matrix mutant by RNA-seq to determine the genes that encode products important to this interaction.

R - As mentioned above, we have studied the relevance of the T6SS in a 3610 context and we have incorporated the data in the new manuscript version. Firstly, we have documented the interaction between the T6SS mutant and wildtype 3610 with images and CFU/ml counts (Suppl. Fig. 19), which shows no difference with respect to the wildtype PCL1606 strain. Secondly, as suggested by reviewer (see below in Page 3 of our Response to Referees), new RNAseq data obtained from the PCL1606 – 3610 interaction have shown differential expression of some genes related to T6SS at 72 h of interaction, when PCL1606 and 3610 are in contact at their colony leading edges. To analyze the effect at a cellular level we have used the methodology developed for the timelapse assays, measuring T6SS promoter expression using PCL1606 carrying a P_{T6SS} Ds-Red construct at 72h during interactions between both 3610 and Δ matrix. We find that T6SS is much more weakly induced during interaction with wildtype 3610 than in the case of the Δ matrix (new Fig. 7). These new results are described in the Results section (Lines 343-349).

In addition, as mentioned above, the corrected graphs now included as Fig. 7 allow us to assign a role for T6SS in the induction of sporulation in *Bacillus* colonies (See Fig 2b and Fig. 7b). These results concur with those obtained in plants, where we see a role for T6SS in the increased sporulation of *Bacillus* strains (See Figures 9 and 10).

Furthermore, we fully agree that a transcriptomic comparison between PCL1606-3610 and PCL1606- Δ matrix interactions will shed light in the understanding of the changes that occur directly related with the absence of extracellular matrix and those dependent on the interaction. We have performed this analysis, and we have found quite interesting results. However, due to the large amount of new information obtained we feel that this data would be better communicated as part of a new manuscript.

[Redacted]

Q - c. The microscopy visualizing the *B. subtilis* motility reporter does not seem relevant to the story (Fig. 6). It is again in a *B. subtilis* matrix mutant background and no biological controls are shown. A more relevant question to pursue would be why the extracellular matrix mediates resistance to *P. chloroaphis* invasion, which could follow after Figure 3.

R - To address these points, we have performed new experiments to determine if the expression of motility-related genes could constitute an escape strategy. We find that increased motility gene expression is related with contact with PCL1606 and is not merely due to loss of the extracellular matrix. In addition, new RNAseq data obtained from the PCL1606-3610 interaction also shows the induction of motility and flagellar related genes in the intermediate area. Furthermore, our RNAseq data comparing 3610 and Δ matrix single colonies show an induction of motility in Δ matrix (See Table 1 in our reply to Reviewer 1), supporting our hypothesis. However, given that this experiment is not part of the paper's core focus on the extracellular matrix, we propose removing explicit references to the existence of a *Bacillus* escape mechanism, while retaining references to changes in motility-related gene expression with modifications to Results (Lines 368-389).

We have also attempted to better characterize the biophysical properties in the presence and absence of the matrix genes to see if they correlate with their altered resistance to invasion. By performing high resolution Magnetic Resonance Imaging and solid state NMR we measured a higher water diffusion rate in Δ matrix compared to 3610, a parameter inversely proportional to bacterial colony compaction, especially in the front line of colony. We also found that the Δ matrix colony is more homogeneous than 3610 in terms of water availability, making the Δ matrix colony fluid. In addition, solid state NMR data support MRI data at the atomic level and highlight differences in the composition of the colonies. While these results do not exclude the involvement of other factors, they strongly suggest a strong correlation between these altered properties and invasion susceptibility.

The new data has been included in the Result section (lines 217-261), Discussion (lines 468-473) and Fig. 4.

Q - 3. There were multiple issues with the Figures and how the data was presented.

- Figure 1: It's difficult to observe the invasion in these images. Can you add an inset with a close-up of the intermediate? Also, please label the colonies directly rather than just with "a,b,c,d,e" – it makes unnecessary work for your reader to have to constantly look back at the legend to figure out what strains are being shown and to interpret the data – simply put the labels on the image.

R - In order to improve consistency between figures and to make the progression of the bacterial interactions easier to follow, we have taken new pictures using a stereo microscope with scale bars included for all images. In addition, we have added labels in the images and magnifications of the intermediate areas. Figure 1 now shows the interactions and the close-up of the intermediates, while control colonies are in Suppl. Fig 3.

Q - • Figure 2: What are a, b, c? Are they I, II, III? Not explained anywhere. Also, a,b,c appear to be identical data to d,e,f – why are both being shown?

R – We have modified and reorganized Fig. 2, 7, Suppl. Fig. 5 and 19 to make them easier to follow. The figures have been divided in three panels (a, b and c), each of them representing a section of the interaction (BA, IA, PA). The section names have been included in each panel. In addition, color legends have been also included in the figures. Readers can find a scheme of the interactions and the sections in (d).

In addition, the raw data shown in the upper parts and in the lower parts of the panels are essentially the same but they were processed in different ways and, in our opinion, they provide interesting and complementary information. The upper part of panels a, b and c shows percentages of populations and *Bacillus* sporulation rates in each of the sections while lower parts of the panels shows CFU/ml counts. Results are complementary as the upper part permits the analysis of the invasion dynamics while the lower parts provide information about the real size of the populations at each time-point.

Q - • Figure 4:

a. See above. It seems to me that the *B. subtilis* matrix mutant would be producing less surfactin during the interaction because less cells *B. subtilis* cells are present. Or is surfactin truly downregulated by *B. subtilis* in the interaction?

R – As stated previously, we have changed the MALDI-TOF MSI representation and now data are presented as heatmaps instead of single-color patterns. In agreement with our data, the number of *B. subtilis* cells present in the interaction is quite constant during the interaction. Furthermore, as shown in Fig. 2, the *Bacillus* population is highly sporulated in the presence of PCL1606. LC-MS/MS data included in Suppl. Figure 15 comparing surfactin production in both interactions suggest similar surfactin production. Thus, in our opinion, the most significant changes are related to the spatial distribution showed by MALDI-TOF MSI rather than the overall amount of surfactin produced.

Q - b. Does PCL 1606 produce a molecule with surfactant properties that may have a different m/z? There are many lipopeptides generated by pseudomonads that might have similar chemical characteristics in terms of facilitating cell spreading

R - LC-MS/MS data obtained does not show any known surfactant produced by PCL1606. In addition, drop collapse assays performed in our group did not show evidence of surfactant production by PCL1606. Moreover, the new Δ matrix Δ surf results presented in this manuscript, as stated above, support the involvement of surfactin in PCL1606 cell spreading.

Q - c. Panel f – what the two colors correspond to in terms of strains did not appear to be mentioned anywhere.

R - Color correspondence is indicated in the figure legend. We have also included a new color correspondence key to the figure.

Q - • Figure 5: Panels b – g are uninterpretable. I have no idea what I am looking at in these images. Some context or orientation for the reader is necessary.

R - Old Fig. 5 (now Fig. 6) has been comprehensively modified and reorganized to make it easier to follow. The new images have been taken using the methodology developed for the timelapse assays. Panels b, c and d show images of: (b) PCL1606 P_{T6SS}-DsRed colony growing alone in LB media, (c) interaction 3610 (CFP) vs PCL1606 P_{T6SS}-DsRed, and (d) Δ matrix (CFP) vs PCL1606 P_{T6SS}-DsRed.

Q - • Figure 6: The microscopy with the motility reporter does not seem relevant to the story, as noted above.

R – As explained above, we have performed new experiments to better test whether the expression of motility-related genes constitutes an escape strategy. Based on this data and the referee's comments we have decided to removing explicit references to the existence of a *Bacillus* escape mechanism, while retaining references to changes in motility-related gene expression with modifications.

Q - • Figure 7cd: Indicate leaf surface. (Again, provide some context in terms of a DIC/brightfield image/diagram/etc. to explain what this image is showing.) Does the white color indicate overlap between strains? Again, this is not mentioned.

R – We have modified the figure (now Figure 8cd) to make the relative positions of the leaf and bacterial layers much clearer. The white color is the consequence of combined cyan and red colors used to indicate emissions CFP and dsRed labelled bacteria rather than any specific labelling or image processing. This results from the naturally poorer resolution in the Z axis projections (a virtual slice through several XY images), which means emissions from groups of neighboring bacteria can appear mixed together. This is a technical artifact rather than real co-localization as this effect is not observed in the original XY images. This effect has been mentioned in the figure legend to avoid any confusion.

Q - 4. I have serious reservations about all of the CFU data. From what I can tell from the methods, the cells were merely "completely resuspended" (line 558) rather than being sonicated prior to plating for CFU counts. It is well established that *B. subtilis* frequently forms cell chains, and sonication is required to separate these chains into individual cells. In the absence of this experimental step, all of the CFU counts could be a dramatic underestimation (i.e. a single chain with many dozens or hundreds of cells would lead to a single apparent CFU; thus the CFU quantification may not accurately represent CFU from single cells.)

R – The reviewer is absolutely correct. It was our mistake for omitting the sonication protocol from the Material and Methods of the original article. All CFU experiments were done by sonicating samples. We have modified the text accordingly. See lines 606-609 in Material and Methods.

Q - Minor comments:

- Keep consistent with strain names (i.e. choose one to use after introducing the strain). There are multiple places where *Pseudomonas* and *P. chloraphis* are used interchangeably with PCL 1606.

R – Thank you. We have modified the text to keep consistency throughout the text.

Q - • Line 48: Change "metabolic consortia" to "metabolic exchange"

- Line 52: Change "they" to "it"

- Line 58-59: Change "little is known about their functional importance in real environments" to "little is known about the functional importance of extracellular matrix components in natural environments"

- Line 77: Change "cohabit" to "coexist"

- Lines 82-96: Changing between past and present tense

- Line 94: Change co-bacterizations to co-colonization of bacteria

- Line 96: Remove determinant

- Line 96: Change "plant seeds germination" to germination of plant seeds

- Line 100: Write out *Pseudomonas*

R – All the changes have been done according to the reviewer suggestions.

Q - • Lines 101-109 belong in the Introduction

R - We have modified last part of the Introduction to incorporate the information provided at the beginning of the results section.

Q - • Figure 1: It's difficult to observe the invasion in these images. Can you add an inset with a close-up of the intermediate?

R - As mentioned above, we have taken new pictures using a stereo microscope with scale bars included for all images. In addition, we have added labels in the images and magnifications of the intermediate areas. Figure 1 now shows the interactions and the close-up of the intermediates, while control colonies are in Suppl. Fig 3.

Q - • Supp Figure 1d. No wrinkles present on *B. subtilis* 3610 after 72h growth on MSgg. What temperature are these grown at?

R – New images have been taken and now 3610 wrinkles can be observed properly. Experiments were performed in LB medium a 28 °C.

Q- • Line 131: Change "bsla" to "bslA"

R – Done.

Q- • Line 154: Remove "in the interaction between the two wild-type strains" because it led me to think you were talking about the intermediate area.

R - We have modified the sentence to avoid confusion.

Q - • Lines 164-165: List percentages of sporulation in the same order for an easier comparison for your reader

• Line 168: Remove "secondary"

• Line 199: Remove "very"

• Line 226: Replace "To clarify this moment of the interaction" with "To gain insight into factors that mediate the observed colony expansion rates"

R - All these comments-suggestions have been modified and included in the new manuscript version.

Q - • Figure 4: Does PCL 1606 produce a molecule with surfactant properties that may have a different m/z?

R - As mentioned above, according to the data obtained by LC-MS/MS, we are not able to detect known surfactants that might be produced by PCL1606. In addition, drop collapse assays performed previously by our group members do not indicate the production of surfactants.

Q - • Line 319: Change "spo0A" to "Spo0A"

• Line 320: Change "interaction" to "intermediate" for consistency

• Line 346: Correct "give rises" to "give rise"

R - All these changes have been included.

Q - • Figure 7cd: Indicate leaf surface. Does the white color indicate overlap between strains?

R - We have indicated the leaf surface. White color is indeed the overlap between the two strains although as indicated above it is technical artifact caused by the poorer resolution obtained in the axial (z) plane by optical microscopy, as the bacterial are close (and presumably in direct contact) but not actually co-localized when examined on a normal lateral (XY) image with a high resolution microscope.

Q - • Line 347: Define "radicle" for those unfamiliar with this plant biology term

R - We have included a definition in the manuscript to clarify this concept. See line 400.

Q - • Lines 411-414: Rewrite sentence in parallel "More specifically, BslA and exopolysaccharide contribute hydrophobicity and resistance to diffusible active molecules, respectively, to *B. subtilis* biofilms

- Line 458: Remove "would be inflicted by" and replace with "is"

- Line 461: Replace "proven" with "shown"

- Line 453: Replace "to be depended on" with "seems to depend on"

- Lines 455-456: Remove "and the maintaining of a small sub-population able to move and escape from the *Pseudomonas* presence by the activation of motility elements." If removing data

R - Thank you for the suggestions. Changes have been incorporated into the main text.

Q - • Line 646: When was LC-MS/MS used in the study?

R - We have used LC-MS/MS to analyze, by molecular networking, the cluster of surfactins produced and to provide additional support, with complementary information, the MALDI-TOF MSI findings. An example of the results obtained by LC-MS/MS can be found in Suppl. Fig. 15.

Q - • Line 1006: State what color line represents *B. subtilis* and PCL 1606.

R - We have followed the same color code during all the figure. This was indicated at the beginning of Fig. 3 legend and now it has been incorporated to other sub-sections of the figure legend. In addition, we have added labels in the images.

Q - • Line 1009: Unclear what the term spatially distributed means:

R - MALDI-TOF MSI data shows the distribution of a detected ion, which corresponds to a detected metabolite, in a two dimensional space, in this case, the layer of culture media containing the microbial interactions (or their monoculture) removed from the Petri dish and place it into the MALDI plate. In this case, we wanted to indicate that surfactins are distributed in agar medium and not only in *Bacillus* colony. We have modified the Fig. 5 title to clarify this.

Q - • Line 1021: Add that *B. subtilis* is labeled using CFP to figure legend

R - We have rewritten figure legends to add more clarity to the figures.

Reviewer #3 (Remarks to the Author):

[The editors asked this reviewer to comment in particular on the MALDI-IMS imaging analyses.]

Overall, the manuscript is clearly written. The work is easy to understand, even for others in a different field. In this report, the authors explored the function significance of biofilms during the interactions between 3610 and PCL 1606. This study demonstrated how the extracellular matrix protects *B. subtilis* colonies from the infiltration by *Pseudomonas*, using microscopy, RNA-seq analysis, and mass spectrometry imaging.

R - We really appreciate comments and suggestions done by reviewer 3. We have tried to take on board their suggestions and believe that the revised submission has been greatly improved as a result. Below we list our responses to the specific comments of the referee.

Q - Specific comments:

In supplemental figure 1, the existence of a "clear" halo is not evident, specifically in (c). Can you add arrows or labels indicating the halo for this image?

R - Done. We have added arrows in Suppl. Fig 1a, b and d.

Q - In line 142-146: It is difficult to find the labels I and II indicated in Fig. 2, 3, and Suppl. Fig 3. Can you add more description to your text (i.e. Fig. 2g-l) so it can be easy to find?

R - Following recommendations made by the Reviewer 2, we have modified the Fig. 2, Fig. 7, Suppl. Fig. 5 and 19 to make them easier to follow. In this case, we have organized the figures in three panels (a, b and c), each of them representing a section of the interaction, which is now indicated in the upper part of each panel and also indicated in the images of the panel d. In addition, we have added a figure legend in the lower part of the panels to make the figure easy to follow.

Q - Consistency: In the caption of Suppl. Fig 3., the text compares an empty bar with a black bar, while in caption Suppl. Fig 4, the text compares a white bar vs. a black bar. Make sure to keep the text and explanation consistent. The same occurs with Fig. 5.

R - Thank you. We have made the suggested changes.

Q - For Fig 3: What is the scale of your white scale bar?

R - Scale is 100 μm . Information have included in the figure legend.

Q - MSI related comments

Is there a reason why you are not including metabolites produced by *B. subtilis* that show similar distribution patterns in both interactions and monocultures in your Suppl. Fig. 5? Show some MS images of metabolites that actually have signal.

R - Thank you for the suggestion. We have included the MS profiles of the single colonies and interactions in Suppl. Fig. 9, 10 and 11. In addition, we have also included distribution patterns of other molecules produced by PCL1606 (See Suppl. Fig 12) and by *Bacillus* strains (See Suppl. Fig 13).

Q - Is there a reason why you are only showing $[M+Na]^+$ MS images, while the molecular network results are of $[M+H]^+$? How does the data compare between the different adducts? Do you observe other adducts in the MS data (i.e. K^+)? Have you tried imaging the summed adduct intensity and doing the same for the molecular network?

R - The main reason to show the MALDI-TOF MSI images of the surfactins was based on the intensity of the detected ion. From MALDI-TOF MSI it was favored the adduct formation of surfactins, while from the LC-MS/MS acquisition, the ESI condition favored the protonated form and we found more surfactin isoforms in the protonated form than with sodium adducts. In the case of MALDI-TOF MSI it is possible to sum the adduct intensity when multiple adducts are detected by simply overlapping all the detected adducts of one metabolite since all of them will have the same spatial distribution. Since it was not the case with surfactins, when only $[M+Na]^+$ were detected by MALDI-TOF MSI, then only these adducts were visualized. In the case of LC-MS/MS data, there are pre-processing tools in development to merge these data (when multiple adducts are detected), however in this study we have mainly considered the presence of protonated ions since their fragmentation spectra was adequate for molecular networks and consequently identification.

Q - How did you generate and compare the MS images? For example, are you generating the image based on signal intensity, ratio to internal standard, TIC, etc?

R - MALDI-TOF IMS data were analyzed using FlexImaging 3.0 (Bruker Daltonics) software. The acquired spectra were normalized by dividing all the spectra by the mean of all data points (TIC normalization method). The resulting mass spectrum was filtered manually in 0.25% (3.0 Da) increments assigning colors to the selected ions associated to the metabolites of interest. This paragraph has been added to the corresponding methods section "Matrix-assisted laser desorption ionization imaging mass spectrometry (MALDI-TOF MSI)", (See lines 708-711).

Q - In line 278-280, the text indicates that the main transcriptional changes can be categorized in (i) amino acid and carbon metabolism. In the MS data, did you see any metabolites that correspond to the amino acid and carbon metabolism?

R - We agree that the correlation between metabolomics and transcriptomics would be very useful to understand the interaction. We have detected some amino acids in the MS data, however is difficult to directly correlate the transcriptional changes and the metabolic profile as these amino acids or products involved in carbon metabolism are intermediaries in the synthesis of other compounds and, therefore, metabolite concentrations might not be directly related with transcriptional expression.

Q - How did you confirm the identity of C14-surfactin and C15-surfactin peaks in MS?

R - The identification of surfactins was done using the calculated molecular formula based on the detected accurate mass (MS1) and the match of their fragmentation pattern (MS2) to annotated fragmentation spectra from GNPS libraries using the molecular network workflow. The mirror plots of both spectra, for C14 and C15 surfactin isoforms, have been extracted from GNPS and included in the supplementary information (See Suppl. Fig. 14b and c).

Q - In line 637, you state that CHCA/DHB is a universal matrix. Be careful when claiming a matrix as "universal" since that does not exist.

R – Thank you for the comment. The “Universal Matrix” commercialized by Sigma-Aldrich (or Fluka), containing 1:1 CHCA:DHB, is no longer commercialized. We removed this name and referred to this matrix as “1:1 mixture of 2,5-dihydroxybenzoic acid and α -cyano-4-hydroxycinnamic acid” in the main text of this manuscript. See lines 701-702.

Reviewers' comments:

Reviewer #1 (Remarks to the Author):

The authors have answered most of my and Reviewer 2 comments, though few experimental and mostly textual concerns remained:

The new sporulation results do not add to much new information. It is well known that kinA and kinB are mainly involved in initiating sporulation. Therefore deletion of kinB is not surprisingly decrease the amount of spores, independently from the competitor. However, kinA was surprisingly not tested in the rebuttal and must be added.

I do not understand what this means: "sporulation levels during interaction with PCL1606 falling to single colony levels" does this mean that sporulation drops to a level that is comparable to pure *B. subtilis* colonies? Please extend to increase understanding.

The statement that "Thus, our data shows that *B. subtilis* responds to the interaction with PCL1606 by activating sporulation even when a functional extracellular matrix prevents colony invasion. " is not valid, as the response to PCL is not mechanistically demonstrated, it should be corrected to "shows that *B. subtilis* activates sporulation even when a functional extracellular matrix prevents colony invasion by PCL1606. "

Line 244: correct "that" to "than"

Again, this statement is not supported by the data: "suggest a role for the T6SS in the maintaining of the *Bacillus* sporulation levels". I understand that author also extend this by a half sentence that this might be due to indirect effects, but other than sporulation rate is different with and without T6SS, no other direct mechanism is presented. This concluding sentence should be strongly reduced for statements, i.e. do not state that T6SS has a role in maintaining sporulation level, but removing T6SS in PCL1606 alters sporulation in *B. subtilis*. I understand that the change is subtle, but the current version suggests results that are not present (i.e. direct involvement) and therefore misleading.

Line 372: "prophage activation" suggests phage assembly and release from the prophages, but this is not examined here. Simply alter and state that transcription of prophage genes is altered. It does not necessarily mean prophage activation. (note the limit of title characters at Nat Comms, as you might need to reduce before final acceptance - friendly suggestion from my previous experience)

Line 392 (sorry I missed this before): please correct "attempts by non-sporulating cells to survive" so it is less anthropomorphic, it is not an attempt, it is a physiological response by the bacterium

Line 509: this sentence is still not supported "However, this is the first demonstration that the T6SS can also have an impact against Gram-positive bacteria by triggering sporulation". There is no proof presented here on the triggering effect. That can be stated is that "However, this is the first indication that T6SS related factors influence sporulation in Gram-positive bacteria"

Line 515: again, this statement is not demonstrated mechanistically: "The first mechanism is KinB-mediated and is dependent on the nutrient availability and environmental changes detected by *Bacillus* strains in the presence of PCL1606 (Suppl. Fig. 7)"

KinB affects the initiation sporulation, but this might be in an indirect additional factor (there is still sporulation happening, right?); nutrient availability, though strongly suggested (and therefore can be hypothesized, but not stated strongly, like in line 524) is not examined here. Does altering nutrient availability change the spore number? If such an experiment is included, nutrient availability could be stated.

Line 528: remove the “fundamental” term, otherwise the sentence is good, as it does not state, but hypothesizes

Line 534: this sentence suggest that T6SS evolved to trigger sporulation and therefore wrong “The T6SS of PCL1606 seems to have adopted a role in these scenarios as a trigger of sporulation” (see also comments above)

548: LB do stand for Lysogeny Broth, as described by Bertani:
<https://jb.asm.org/content/186/3/595>

Fig 7 title is not correct, no direct proof of T6SS triggering sporulation (see above)

Please revise Fig 9 title: T6SS does not kills sporulation, also triggering effect is not demonstrated as highlighted above

Fig S6: i do not understand how can sporulation rate be higher than 100%, this suggest mistakes in the assay: higher CFU measured after heat inactivation than before?

Reviewer #3 (Remarks to the Author):

All of my comments and concerns have been addressed

Reviewers' comments:

Reviewer #1 (Remarks to the Author):

The authors have answered most of my and Reviewer 2 comments, though few experimental and mostly textual concerns remained:

Q - The new sporulation results do not add to much new information. It is well known that kinA and kinB are mainly involved in initiating sporulation. Therefore deletion of kinB is not surprisingly decrease the amount of spores, independently from the competitor. However, kinA was surprisingly not tested in the rebuttal and must be added.

R – We have included a kinA mutant experiment to the revised manuscript (See Suppl. Fig. 7 and Lines 163-169). As observed for the kinB mutant, similar sporulation levels were obtained in kinA mutant background in both the presence and absence of PCL1606. From this we suggest the involvement of both kinases in mediating *Bacillus* sporulation induced by PCL1606 (please, see also our response to the next comment).

Q - I do not understand what this means: “sporulation levels during interaction with PCL1606 falling to single colony levels” does this mean that sporulation drops to a level that is comparable to pure *B. subtilis* colonies? Please extend to increase understanding.

R – We have modified and clarified the sentence. Our results now demonstrate that 3610 WT and 3610 mutants for kinB or kinA show similar sporulation levels in the absence of PCL1606 (See Sup. Fig. 6A, 72h and Supp. Fig 7). The presence of PCL1606 triggers sporulation in the wild-type 3610 strain (as in Δ kinC or Δ kinD) but not in Δ kinA or Δ kinB, suggesting that the two histidine kinases are involved in sensing the signals that trigger sporulation in the interaction between *B. subtilis* and PCL1606, when cells are not in contact.

Q - The statement that “Thus, our data shows that *B. subtilis* responds to the interaction with PCL1606 by activating sporulation even when a functional extracellular matrix prevents colony invasion. “ is not valid, as the response to PCL is not mechanistically demonstrated, it should be corrected to “shows that *B. subtilis* activates sporulation even when a functional extracellular matrix prevents colony invasion by PCL1606. “

R – The reviewer is right. We have modified the statement to make it clearer. Thank you.

Q - Line 244: correct “that” to “than”

R - Done.

Q - Again, this statement is not supported by the data: “suggest a role for the T6SS in the maintaining of the Bacillus sporulation levels”. I understand that author also extend this by a half sentence that this might be due to indirect effects, but other than sporulation rate is different with and without T6SS, no other direct mechanism is presented. This concluding sentence should be strongly reduced for statements, i.e. do not state that T6SS has a role in maintaining sporulation level, but removing T6SS in PCL1606 alters sporulation in B.subtilis. I understand that the change is subtle, but the current version suggests results that are not present (i.e. direct involvement) and therefore misleading.

R – The statement was certainly not clearly presented in the previous version of the manuscript. We have modified the sentence according to the suggestions proposed. See lines 362-363.

Q - Line 372: “prophage activation” suggests phage assembly and release from the prophages, but this is not examined here. Simply alter and state that transcription of prophage genes is altered. It does not necessarily mean prophage activation. (note the limit of title characters at Nat Comms, as you might need to reduce before final acceptance - friendly suggestion from my previous experience)

R – Thank you. We have modified the section title as prophage activation has not been studied, although differential expression of prophage genes was found in transcriptomic analyses.

Q - Line 392 (sorry I missed this before): please correct “attempts by non-sporulating cells to survive” so it is less anthropomorphic, it is not an attempt, it is a physiological response by the bacterium

R - Modified, thank you.

Q - Line 509: this sentence is still not supported “However, this is the first demonstration that the T6SS can also have an impact against Gram-positive bacteria by triggering sporulation”. There is no proof presented here on the triggering effect. That can be stated is that “However, this is the first indication that T6SS related factors influence sporulation in Gram-positive bacteria”

R - The sentence has been modified according to the reviewer indications.

Q - Line 515: again, this statement is not demonstrated mechanistically: “The first mechanism is KinB-mediated and is dependent on the nutrient availability and environmental changes detected by Bacillus strains in the presence of PCL1606 (Suppl. Fig. 7)”

KinB affects the initiation sporulation, but this might be in an indirect additional factor (there is still sporulation happening, right?); nutrient availability, though strongly suggested (and therefore can be hypothesized, but not stated strongly, like in line 524) is not examined here. Does altering

nutrient availability change the spore number? If such an experiment is included, nutrient availability could be stated.

R – We agree with the reviewer and have modified the text in order to weaken the statement and present it as a hypothesis/suggestion instead.

Q - Line 528: remove the “fundamental” term, otherwise the sentence is good, as it does not state, but hypothesizes

R – Done.

Q - Line 534: this sentence suggest that T6SS evolved to trigger sporulation and therefore wrong “The T6SS of PCL1606 seems to have adopted a role in these scenarios as a trigger of sporulation” (see also comments above)

R – We have modified the sentence to clarify that. Thank you.

Q - 548: LB do stand for Lysogeny Broth, as described by Bertani: <https://jb.asm.org/content/186/3/595>

R – Reviewer is right. Modified.

Q - Fig 7 title is not correct, no direct proof of T6SS triggering sporulation (see above)

R – Figure 7 title has been modified, thank you.

Q - Please revise Fig 9 title: T6SS does not kills sporulation, also triggering effect is not demonstrated as highlighted above

R – We have modified the figure title. Thank you.

Q - Fig S6: i do not understand how can sporulation rate be higher than 100%, this suggest mistakes in the assay: higher CFU measured after heat inactivation than before?

R – The reviewer is right. Standard deviations were wrong in some cases. New replicates have been added to the experiments and incorporated in Suppl. Fig. 6.

Reviewer #3 (Remarks to the Author):

All of my comments and concerns have been addressed